# A mitochondrial pentatricopeptide repeat protein enhances cold tolerance by modulating mitochondrial superoxide in rice

Xiaofeng Zu[1,7], Lilan Luo [ORCID][1,7] ✉, Zhen Wang [ORCID][1,7], Jie Gong[1,2], Chao Yang[1], Yong Wang[3], Chunhui Xu[3], Xinhua Qiao[4], Xian Deng [ORCID][1], Xianwei Song [ORCID][1], Chang Chen[4,5], Bao-Cai Tan[3] & Xiaofeng Cao [ORCID][1,5,6] ✉

Cold stress affects rice growth and productivity. Defects in the plastid-localized pseudouridine synthase OsPUS1 affect chloroplast ribosome biogenesis, leading to low-temperature albino seedlings and accumulation of reactive oxygen species (ROS). Here, we report an *ospus1-1* suppressor, *sop10*. *SOP10* encodes a mitochondria-localized pentatricopeptide repeat protein. Mutations in *SOP10* impair intron splicing of the *nad4* and *nad5* transcripts and decrease RNA editing efficiency of the *nad2*, *nad6*, and *rps4* transcripts, resulting in deficiencies in mitochondrial complex I, thus decrease ROS generation and rescuing the albino phenotype. Overexpression of different compartment-localized superoxide dismutases (SOD) genes in *ospus1-1* reverses the ROS over-accumulation and albino phenotypes to various degrees, with Mn-SOD reversing the best. Mutation of *SOP10* in *indica* rice varieties enhances cold tolerance with lower ROS levels. We find that the mitochondrial superoxide plays a key role in rice cold responses, and identify a mitochondrial superoxide modulating factor, informing efforts to improve rice cold tolerance.

Rice (*Oryza sativa* L.) evolved in tropical and subtropical areas[1–4]; cold stress severely affects rice growth and yield, thereby limiting its global distribution[5]. Using molecular genetic tools to improve rice cold tolerance may help maintain rice production and expand its geographic distribution. Screening for genes involved in cold tolerance and understanding their underlying mechanisms provides important information for such approaches.

Reactive oxygen species (ROS) play key roles in plant growth and development[6], and mediating the response to abiotic and biotic stresses[7]. ROS are produced passively by metabolic pathways or actively in different organelles and subcellular locations, such as chloroplasts, mitochondria, peroxisomes, and the apoplast[7]. The balance between ROS production and scavenging systems tightly regulates ROS levels. ROS accumulate and act as signals in plant vegetative apical meristem development, organogenesis, and stress responses. For example, enrichment of superoxide ($O_2^{\cdot}$) in stem cells activates the *WUSCHEL* gene to maintain stem cell activities, whereas hydrogen peroxide ($H_2O_2$) accumulation in the peripheral zone promotes cell differentiation[8,9]. Under stress conditions, ROS accumulate and regulate gene expression in the nucleus and organelles to activate plant

[1]State Key Laboratory of Plant Genomics and National Center for Plant Gene Research, Institute of Genetics and Developmental Biology, Chinese Academy of Sciences, Beijing 100101, China. [2]The Municipal Key Laboratory of the Molecular Genetics of Hybrid Wheat, Institute of Hybrid Wheat, Beijing Academy of Agriculture and Forestry Sciences, Beijing 100097, China. [3]Key Laboratory of Plant Development and Environmental Adaptation Biology, Ministry of Education, School of Life Sciences, Shandong University, Qingdao 266237, China. [4]National Laboratory of Biomacromolecules, CAS Center for Excellence in Biomacromolecules, Institute of Biophysics, Chinese Academy of Sciences, Beijing 100101, China. [5]University of Chinese Academy of Sciences, Beijing 100049, China. [6]CAS Center for Excellence in Molecular Plant Sciences, Chinese Academy of Sciences, Beijing 100101, China. [7]These authors contributed equally: Xiaofeng Zu, Lilan Luo, Zhen Wang. ✉e-mail: luolilan@genetics.ac.cn; xfcao@genetics.ac.cn

defenses. Cold stress disrupts cellular homeostasis, resulting in the accumulation of ROS, which is closely linked with the cold stress response[10-18]. Identifying key regulators of ROS production and signaling may enable researchers to manipulate ROS to engineer or breed cold-tolerant crops.

ROS include $O_2^{\cdot-}$, $H_2O_2$, singlet oxygen ($^1O_2$), ozone, hydroxyl radical (OH$^{\cdot}$), and organic and inorganic peroxides; each of these ROS interacts with specific targets and signaling pathways[19]. ROS are independently produced and scavenged at different rates in different cellular compartments to keep ROS at basal concentrations and control ROS signaling[7]. The type and level of ROS in different compartments can differ in response to distinct stress or cellular conditions, triggering stress-specific signal transduction pathways that activate stress-specific acclimation and defense mechanisms[20-23]. However, which cellular compartment produces the ROS that affect cold tolerance, and which species of ROS is important in cold responses, remain to be elucidated.

Our previous study identified a mutant of chloroplast-localized pseudouridine synthase, *ospus1-1*, which displayed albino seedlings under low temperature[12]. Dysfunction of OsPUS1 leads to aberrant chloroplast ribosome biogenesis and thus defective chloroplast

development at low temperatures, which disturbs the balance between growth and stress responses and enhances ROS accumulation. However, the detailed mechanism by which OsPUS1 affects ROS homeostasis and cold stress responses requires further investigation.

In this study, we isolated a suppressor of *ospus1-1*; this mutant partially rescues the low temperature–sensitive albino phenotype in *ospus1-1* and affects a mitochondrial pentatricopeptide repeat (PPR) protein. The suppressor impairs the group II intron splicing of the *nad4* and *nad5* transcripts and disturbs RNA editing of the *nad2*, *nad6*, and *rps4* transcripts, resulting in reduced complex I capacity and suppression of ROS accumulation and the albino phenotype. By overexpressing superoxide-scavenging enzymes, we further show that mitochondrial superoxide is essential for the cold response and chloroplast development under extreme conditions. Importantly, we generated *sop10* single mutants in a series of *indica* rice varieties and found that these are cold tolerant, thus producing genetic resources for rice breeding.

## Results
### *SOP10* encodes a mitochondrial PPR protein

To elucidate the molecular mechanism by which *ospus1* produces the low temperature–sensitive albino phenotype, we screened for *suppressor of ospus1* (*sop*) mutants. To this end, we generated an ethyl methanesulfonate (EMS) mutagenized library of *ospus1-1* seeds. By screening M₂ populations for green seedlings at low temperature (22 °C), we obtained a suppressor mutant; the *ospus1-1 sop10-1* plants exhibited green seedlings at 22 °C similar to wild-type 9311 (Fig. 1a). Genetic analyses showed that *sop10-1* is a recessive mutant.

To clone *SOP10*, we performed MutMap sequencing and obtained a single-nucleotide polymorphism (SNP)-index peak in the genomic interval from 3.61 to 3.67 Mb on chromosome 2 (Fig. 1b). Of the ten SNPs identified in the candidate region, SNP-3615690 was in the first exon of the gene LOC_Os02g07050, which encodes a putative mitochondrial PPR protein (Fig. 1c). This SNP is a G1475A point mutation, leading to a G492D amino acid substitution in the protein sequence (Fig. 1c).

To verify that the point mutation in LOC_Os02g07050 caused the suppression of the *ospus1* phenotype, we complemented the suppressor mutant by expressing the LOC_Os02g07050 gene in the *ospus1-1 sop10-1* background. All the transformants with $P_{SOP10}$:*SOP10* (designated *comp*) showed the *ospus1* albino phenotype at 22 °C (Fig. 1a). In addition, we generated new *sop10* mutant alleles (named *sop10^{CR}*) by CRISPR/Cas9-mediated genome editing in the *ospus1-1* background. Three independent *ospus1-1 sop10^{CR}* lines showed green leaves at 22 °C, similar to the wild type (Fig. 1a and Fig. S1). Together, these results show that mutations in LOC_Os02g07050 are responsible for suppressing *ospus1-1* phenotype in *ospus1-1 sop10-1*.

### SOP10 participates in intron splicing of the *nad4* and *nad5* transcripts

Previously, we reported that OsPUS1 was required for the maturation of chloroplast rRNAs at low temperature[12]. Therefore, we wondered whether the defects of chloroplast rRNA processing were rescued in *ospus1-1 sop10-1*. Northern blot analysis showed that the *sop10* mutants accumulated high levels of pre-16S and pre-23S rRNA, indicating that the suppression of the *ospus1-1* phenotype by *sop10* is independent of chloroplast rRNA processing (Fig. S2).

PPR proteins contain tandem repeats of degenerate 35-amino-acid motifs (PPR motifs, P motif for short), each of which forms an antiparallel α-helix[24]. PPR motif variants include the L (long, more than 35 amino acids) and S (short, 31 or fewer amino acids) motifs. Based on the P, L and S motifs, PPR proteins can be divided into P and PLS subfamilies. The rice genome harbors 491 *PPR* genes, of which 246 genes belong to the P subfamily, and 245 genes belong to the PLS subfamily[25]. *SOP10* (LOC_Os02g07050) encodes a PLS-E2 type PPR

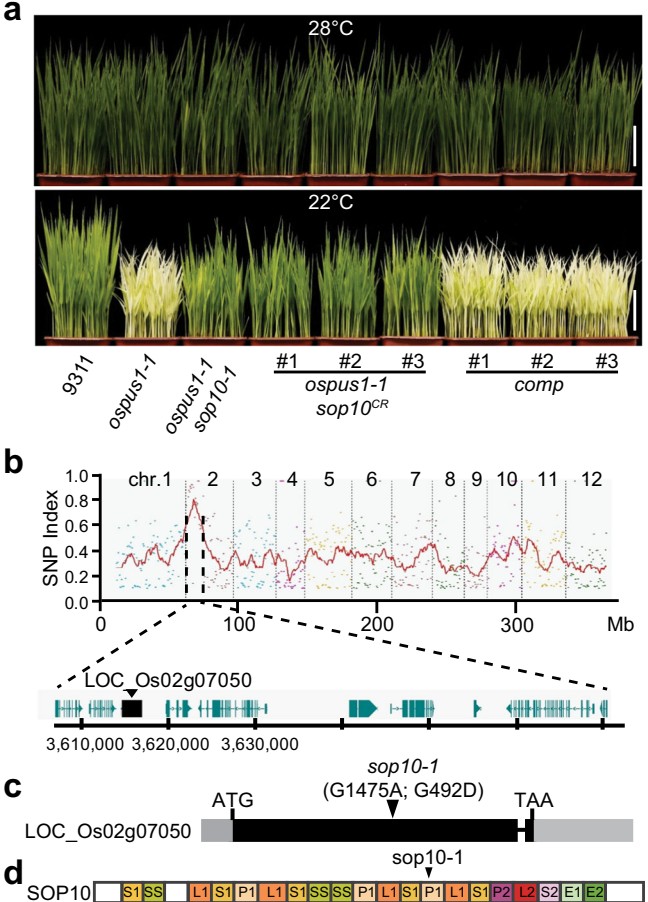

**Fig. 1 | Identification and cloning of *sop10*. a** Phenotypic response of the 9311, *ospus1-1*, *ospus1-1 sop10-1*, *ospus1-1 sop10^{CR}* and *ospus1-1 sop10-1* complementation (*comp*) plants to temperature (28 °C and 22 °C). Scale bar, 5 cm. **b** SNP index of twelve chromosomes plotted by Mutmap sliding window analysis for cloning of *SOP10*. The x-axis represents the positions of twelve chromosomes. The y-axis represents the SNP index. SNP index was created by averaging SNP indices from a moving window spanning five consecutive SNPs and moving the window one SNP at a time. **c** Schematic diagram of *SOP10* gene. The black triangle indicates the mutation site (G1475A; G492D) in *ospus1-1 sop10-1*. **d** Protein structure of OsSOP10 in rice, the PPR motifs were shown in different colored box.

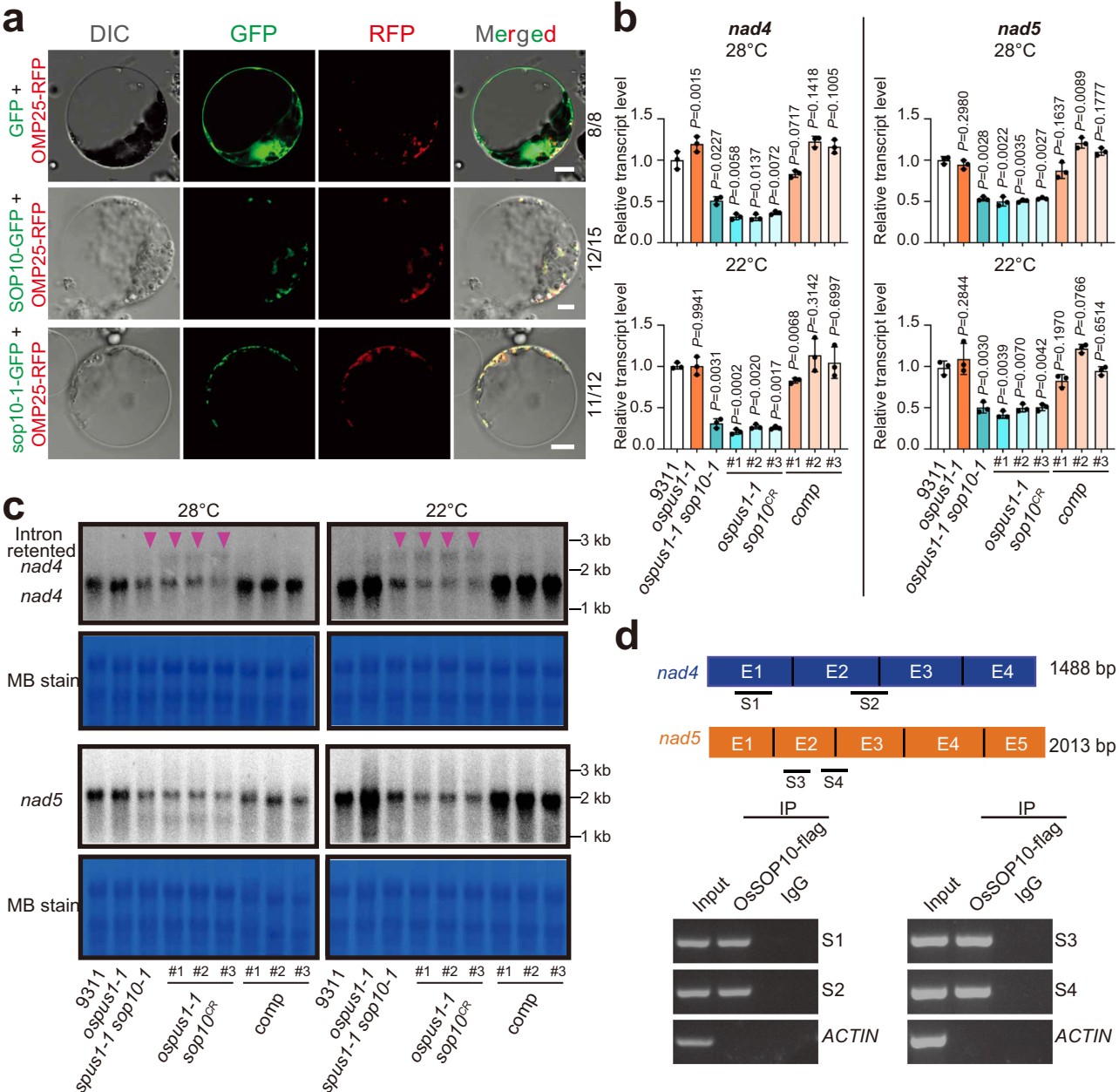

**Fig. 2 | SOP10 binds to the *nad4* and *nad5* transcripts and affects splicing of their first intron. a** Subcellular localization of SOP10 and sop10-1 in rice protoplasts. The mutation in sop10-1 protein did not change its location in mitochondria. OMP25-RFP was used as a mitochondrial marker. The numbers on right side of each panel show how often this representative subcellular localization pattern occurred, relative to total examined cells. Scale bars, 5 μm. **b** Relative expression levels of *nad4* and *nad5* in 9311, *ospus1-1* mutant, suppressors of *ospus1-1* (*ospus1-1 sop10-1* and three *ospus1-1 sop10^CR* lines) and *ospus1-1 sop10-1* complementation (*comp*) plants at 28 °C and 22 °C. The relevant raw data are reported in Supplementary Data 1. Values are means ± S.D. (*n* = 6) of three biological replicates. Differences between *ospus1-1*, *sop10* mutants, complementation lines (*comp*) and wild type are determined by unpaired Student's *t*-test (two-tailed). Exact *P*-values are shown. **c** Northern blot analysis to detect the *nad4* and *nad5* transcripts. Total RNA was

extracted from seedlings of 9311, *ospus1-1*, four suppressors of *ospus1-1*, and *comp* plants grown at 28 °C and 22 °C. The specific fragments of *nad4* and *nad5* used as the probes. Red triangles indicate the intron retention in *nad4*. Methylene blue staining (MB stain) is shown as a loading control. Experiments were repeated three times with similar results. Source data for **c** are provided as a Source Data file. **d** The interaction of SOP10 with *nad4* and *nad5* transcripts demonstrated by RNA immunoprecipitation (RIP). S1 and S2 represent the specific fragments of *nad4* used for RIP RT-PCR; S3 and S4 represent the specific fragments of *nad5* used for RIP RT-PCR. RIP assays were performed with anti-FLAG and anti-IgG antibodies in 9311 and SOP10-FLAG transgenic calli. 0.1% of total RNA was used as input. *ACTIN* was used as the negative control. Experiments were repeated two times independently with similar results.

protein. Phylogenic analysis showed that SOP10 is highly conserved across different plant species (Fig. S3). SOP10 contains 18 PLS-type PPR motifs with C-terminal E1 and E2 motifs predicted by the PPR HMMer profiles[25,26], and the mutated amino acid in sop10-1 is located at the 13^th P1 motif (Fig. 1d), which is conserved among different species (Fig. S3).

To investigate the function of SOP10, we first examined its subcellular localization. The green fluorescence of SOP10-GFP co-localized with the mitochondrial OMP25-RFP marker[27], indicating that the SOP10 is localized in the mitochondria, and the mutated sop10-1 also localizes in mitochondria (Fig. 2a).

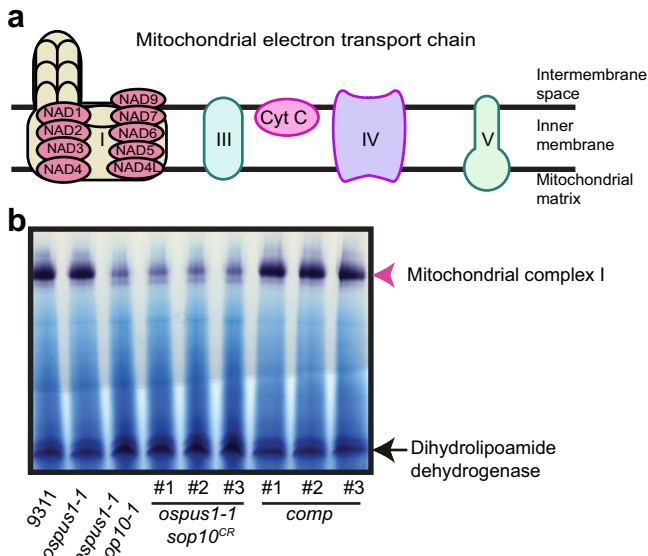

**a** Mitochondrial electron transport chain

**Fig. 3 | Mitochondrial complex I capacity was decreased in *sop10* mutants.**
**a** Diagram of the mitochondrial electron transport chain. The two black lines represent the limits of the membrane bilayer. NADs are core subunits of mitochondrial complex I encoded by the mitochondrial genome. **b** In-gel assay of NADH oxidase capacity. The plants were grown at 22 °C. The dihydrolipoamide dehydrogenase activity was used as a loading control. Experiments were repeated three times independently with similar results. Source data for (**b**) are provided as a Source Data file.

Mitochondrial PPR proteins participate in posttranscriptional processes such as RNA editing[28], splicing[29–41], and stability, to regulate the expression of mitochondria-encoded genes[42–44]. We, therefore, analyzed the levels of all 56 transcripts transcribed from the rice mitochondrial genome by using RT-qPCR in *ospus1-1 sop10* plants (Supplementary Data 1). Among all the mitochondrial transcripts, *nad4* and *nad5* transcripts were remarkably decreased in *ospus1-1 sop10* mutants (*ospus1-1 sop10-1* and *ospus1-1 sop10^{CR}*) and recovered in complemented lines under normal (28 °C) or low temperature (22 °C) growth conditions (Fig. 2b), indicating that SOP10 is involved in regulating *nad4* and *nad5* expression.

This was further confirmed by northern blot analysis (Fig. 2c), which showed a larger band for the *nad4* transcript, suggesting that intron retention may occur (Fig. 2c). Rice *nad4* has four exons with three *cis*-splicing introns and *nad5* has five exons with two *cis*-splicing introns (1 and 4) and two *trans*-splicing introns (2 and 3). Indeed, by amplifying fragments containing each of the introns in the *nad4* and *nad5* transcripts with specific primer pairs (Supplementary Table 1), we found that the first intron *nad4i461g2* and *nad5i230g2* (in group II-type introns, the intron name is composed of the gene name, the letter "i" for intron, and the insertion site in the orthologous gene of *Marchantia polymorpha*[45,46], g2: group 2 intron) was retained, while the other introns in *nad4* and *nad5* were normally spliced in *ospus1-1 sop10-1* plants (Figs. S4, S5).

Since PPR proteins are RNA-binding proteins, we performed RNA immunoprecipitation (RIP) experiments to investigate whether *nad4* and *nad5* are direct targets of SOP10. As shown in Fig. 2d, the *nad4* and *nad5* transcripts were immunoprecipitated by SOP10-FLAG, suggesting that SOP10-FLAG bound to the *nad4* and *nad5* mRNAs[24,47,48]. Taken together, these results indicate that SOP10 directly binds to *nad4* and *nad5* transcripts and participates in the splicing of the mitochondrial introns *nad4i461g2* and *nad5i230g2* in a temperature-independent manner.

## SOP10 is required for C-to-U editing at 5 sites in mitochondria

Since SOP10 is a PLS-E2 type PPR protein, which primarily participates in C-to-U editing[44], we tested whether it participates in editing mitochondrial RNAs. Indeed, strand- and transcript-specific PCR-seq (STS-PCR-seq) detected C-to-U modifications in 34 rice mitochondrial protein-encoding genes[49,50]. The results showed that the C-to-U editing at five target Cs in three mitochondrial transcripts (*nad2*, *nad6*, *rps4*) was severely decreased in the *sop10* mutant compared with the wild type (Fig. S6a, Supplementary Data 2). The editing efficiency of the five sites was reduced by > 20% in the *sop10-2* mutant. We used RT-PCR to amplify the coding sequence (CDS) region and directly sequenced the amplicons in the mutant and the wild type. The results are consistent with the STS-PCR-seq results (Fig. S6b). The editing deficiency at these sites results in amino acid changes in the encoded proteins (*nad2*, P416L, P470S; *nad6*, S53F; *rps4*, P338L). These results indicate that SOP10 is required for C-to-U RNA editing of the *nad2*, *nad6*, and *rps4* transcripts in mitochondria.

## Mitochondrial complex I capacity is decreased in *sop10* mutants

Since NAD2, NAD4, NAD5, and NAD6 are subunits of mitochondrial complex I[51] (Fig. 3a), the deficiency in RNA editing of *nad2* and *nad6*, and the decreased expression of *nad4* and *nad5* may reduce the amount of functional complex I. To test this, we analyzed the NADH oxidase capacity of complex I and found that complex I capacity was significantly lower in *ospus1-1 sop10* mutants (*ospus1-1 sop10-1* and *ospus1-1 sop10^{CR}*) compared with wild type, but was restored to that of the wild type in the complemented lines (Fig. 3b). Further, we generated *sop10^{CR}* single mutants in the Nipponbare background and found that the complex I capacity in *sop10^{CR}*/Nip single mutants was lower than in wild type. This indicates that the function of SOP10 in modulating complex I capacity is independent of the *indica* or *japonica* genetic background (Fig. S7). Altogether, these results suggest that SOP10 alters complex I capacity by decreasing the levels of the *nad4* and *nad5* transcripts and RNA editing of *nad2* and *nad6*, which indicates that complex I affects the low-temperature sensitivity in *ospus1* mutants.

## *sop10* suppresses ROS accumulation in *ospus1-1* mutants

The mitochondrial electron transport chain (mETC) is one of the main sources of ROS production in plants[52–56]. Our data presented above indicate that the low temperature–sensitive albinism in *ospus1-1* requires sufficient amounts of functional complex I. We recently reported that ROS accumulated in *ospus1-1* at low temperature[12]. Therefore, we first examined the ROS levels in suppressors by staining with nitroblue tetrazolium (NBT) and 3,3′-diaminobenzidine (DAB). As shown in Fig. 4a, the NBT staining signal, which indicates the $O_2^{\cdot-}$ accumulation, appears in the leaf blade at high levels in *ospus1-1* leaves at 22 °C but at wild-type levels in *ospus1-1 sop10* mutants and was comparable to that of *ospus1-1* in complemented plants. By contrast, the DAB staining signal, which indicates $H_2O_2$ accumulation, only appeared at the leaf tip not the whole leaf blade in *ospus1-1* at 22 °C. The *sop10* mutation suppressed $H_2O_2$ accumulation in the leaf tip. However, at 28 °C, the ROS levels did not significantly differ in 9311, *ospus1-1*, suppressor mutants, and complemented plants. These results implied that $O_2^{\cdot-}$ and $H_2O_2$ have different roles in the *ospus1-1* low-temperature albino phenotype.

The *ospus1-1* albino phenotype occurs in the whole leaf blade, which is consistent with the accumulation of $O_2^{\cdot-}$ but not $H_2O_2$ (Fig. 4a, b), suggesting that $O_2^{\cdot-}$ may be related to the albino phenotype. We therefore evaluated the activity of superoxide dismutase (SOD), which scavenges superoxide. The rice genome harbors eight *SOD* genes, encoding one mitochondria-localized Mn-SOD, two chloroplast-localized Fe-SODs, and five cytosolic Cu/Zn-SODs (Fig. S8a). Expression of LOC_Os05g25850 encoding Mn-SOD, LOC_Os06g05110

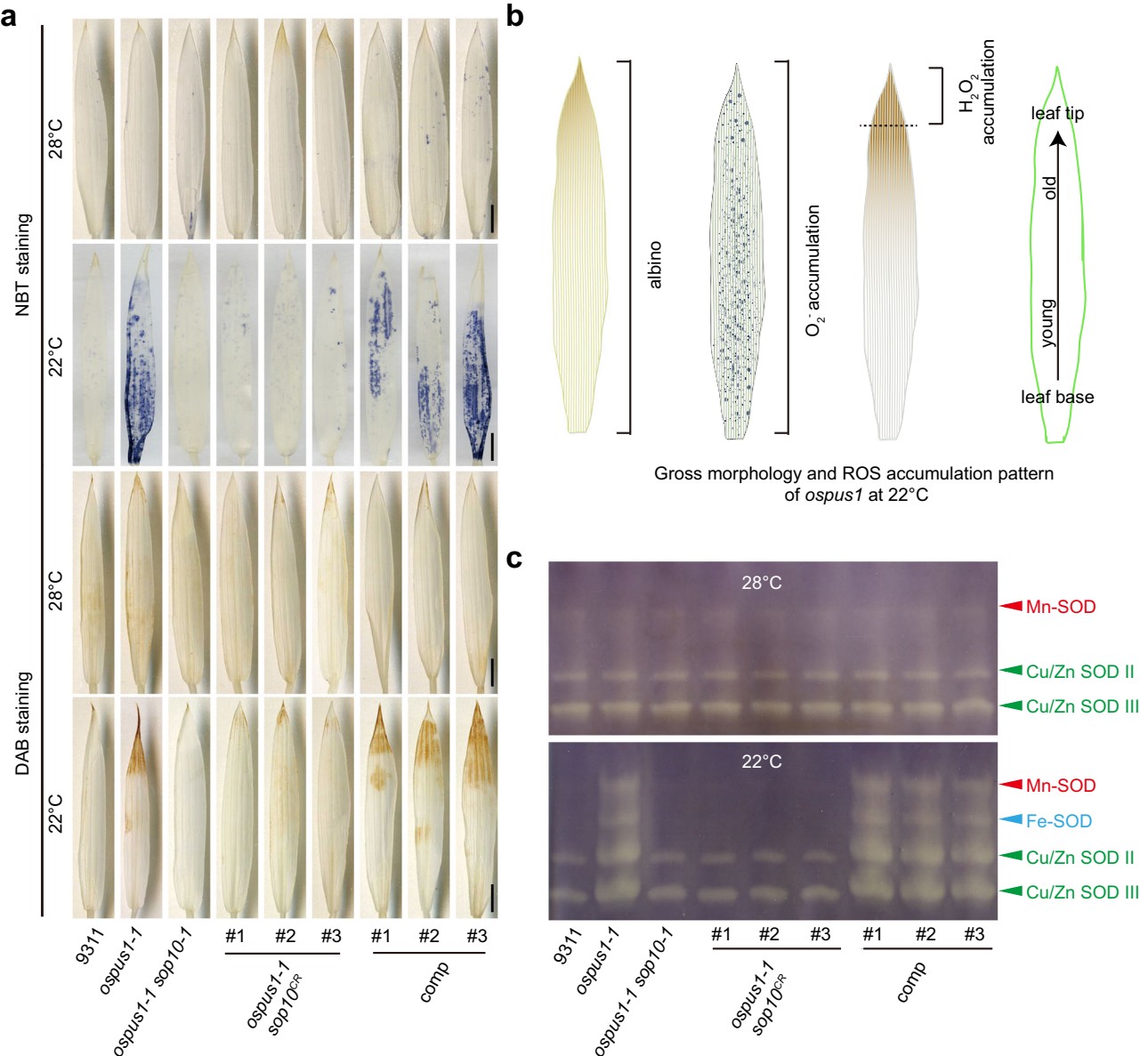

**Fig. 4 | ROS accumulated in *ospus1-1* and the accumulation was suppressed in suppressor mutants. a** NBT and DAB staining were used to assess the accumulation of $O_2^{\cdot-}$ and $H_2O_2$, respectively. Scale bars, 1 cm. **b** Schematic drawing of relationship between albino phenotype and ROS accumulation pattern in *ospus1-1* at 22 °C. The albino occurred in the whole leaf blade which is consistent with $O_2^{\cdot-}$ accumulation pattern. **c** In-gel activity assay of various SOD isozymes in seedlings of wild type, mutants, suppressor mutants, and complemented plants grown at 28 °C or 22 °C. Equal amounts (20 μg) of total protein were subjected to native-PAGE to separate different SOD isozymes. Experiments were repeated three times independently with similar results. Source data for (**a**, **c**) are provided as a Source Data file.

encoding Fe-SOD, and LOC_Os07g46990 encoding Cu/Zn-SOD, was upregulated at 22 °C (Fig. S8b). An in-gel SOD activity assay showed that activities of all four SOD isoforms were significantly higher than wild type in the *ospus1-1* mutant and complemented plants but were similar to wild type in *ospus1-1 sop10* mutants (Fig.4c). Together, these observations suggest that $O_2^{\cdot-}$ accumulated in the leaf blade of *ospus1-1* and induced upregulation of SOD activity; *sop10* suppressed the $O_2^{\cdot-}$ accumulation, probably by inhibiting $O_2^{\cdot-}$ production, and SOD activity was not upregulated.

**Overexpression of SOD genes rescues *ospus1-1* phenotypes**
The ROS staining data indicated a link between $O_2^{\cdot-}$ accumulation and the albino phenotype in *ospus1-1* (Fig. 4a, b). To test whether accumulated $O_2^{\cdot-}$ causes the low temperature–sensitive albino phenotype in *ospus1-1*, we attempted to reduce $O_2^{\cdot-}$ levels by overexpressing SOD,

which can scavenge $O_2^{\cdot-}$, in *ospus1-1*. Based on the *SOD* transcript level and in-gel activity assay, we successfully overexpressed each of the genes encoding SODs that responded to low temperature and localized in mitochondria, chloroplasts, or the cytosol, in *ospus1-1* plants. As shown in Fig. 5a, overexpression of *Mn-SOD* or *Cu/Zn-SOD* resulted in a 2–5-fold elevation of its transcript, and overexpression of *Fe-SOD* resulted in elevation of its transcript by approximately 30-fold in the overexpressing plants grown at 28 °C or 22 °C (Fig. 5a).

Overexpression of the three types of SOD in *ospus1-1* rescued the albino phenotype of *ospus1-1* under low temperature to varying degrees. As shown in Fig. 5, overexpression of a mitochondria-localized Mn-SOD mostly rescued the *ospus1-1* phenotype, but cytosol-localized Cu/Zn-SOD rescued to a lesser extent, and chloroplast-localized Fe-SOD only slightly rescued the *ospus1-1* phenotype (Fig. 5b). We also examined the cellular $O_2^{\cdot-}$ and $H_2O_2$ levels in transgenic plants and found that

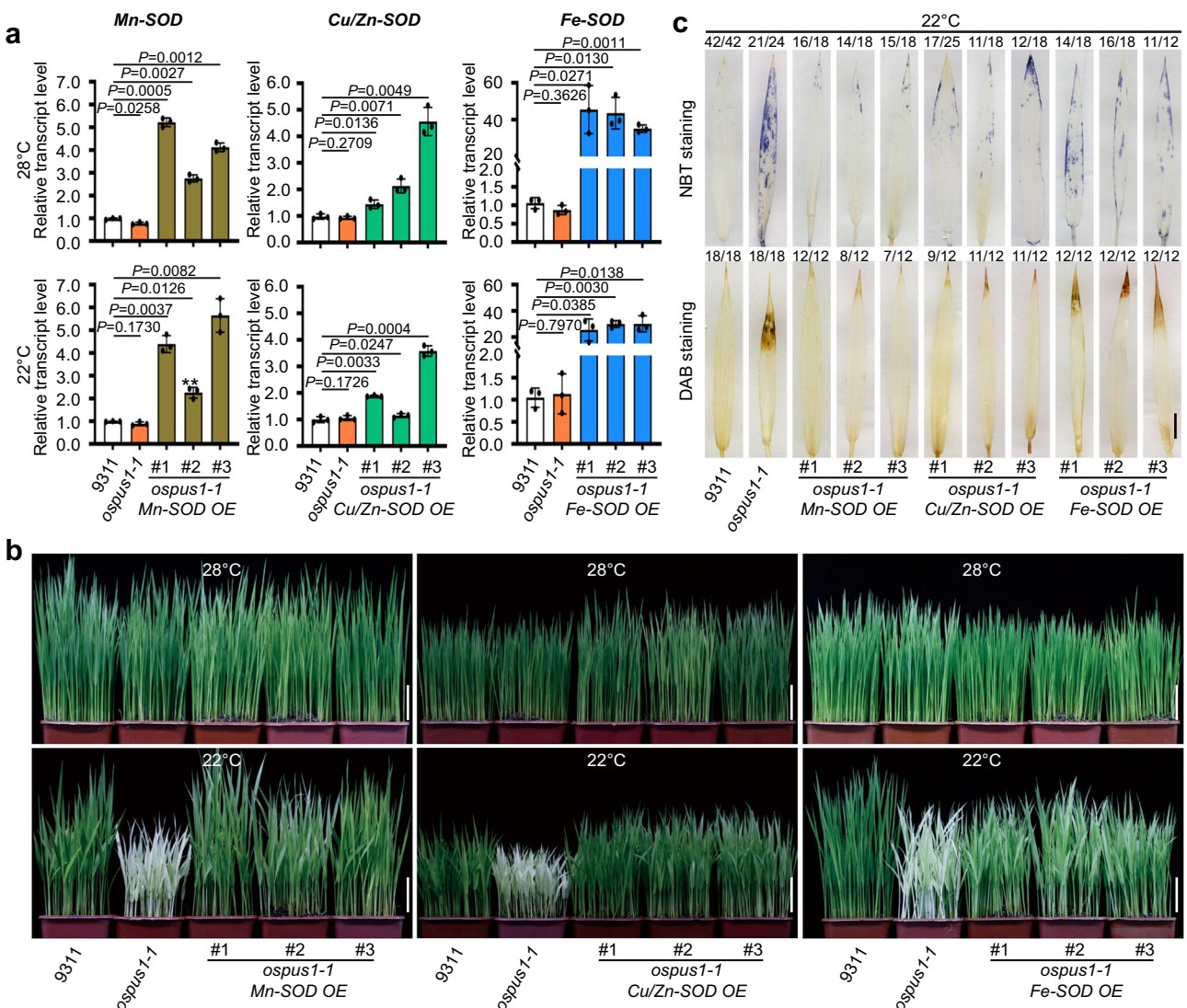

**Fig. 5 | Overexpression of SOD genes suppresses ROS accumulation and the albino phenotype in os*pus1-1*. a** Relative expression levels of *Mn-SOD* (LOC_Os05g25850), *Cu/Zn-SOD* (LOC_Os07g46990) and *Fe-SOD* (LOC_Os06g05110) genes in 9311, *ospus1-1* mutant, and the SOD overexpression plants grown at 28 °C or 22 °C. Values are means ± S.D. (*n* = 6) of three biological replicates. Differences between *ospus1-1* mutant and SOD overexpression lines grown under the same conditions are determined by unpaired Student's *t*-test (two-tailed). Exact *P* values are shown. **b** Phenotypic responses to temperature (28 °C or 22 °C) of the 9311, os*pus1-1*, and os*pus1-1* transgenic plants overexpressing *Mn-SOD* (LOC_Os05g25850), *Fe-SOD* (LOC_Os06g05110), or *Cu/Zn-SOD* (LOC_Os07g46990). Scale bars, 5 cm. **c** Accumulated ROS in *ospus1-1* was suppressed by overexpressing SOD-encoding genes. The accumulation of $O_2^{·-}$ and $H_2O_2$ in the leaf of rice grown at 22 °C in (**b**) were assessed by NBT and DAB staining, respectively. The numbers on top of each photo show how often this representative staining pattern occurred, relative to total examined pictures. Scale bars, 1 cm. Source data for (**a**–**c**) are provided as a Source Data file.

the cellular $O_2^{·-}$ and $H_2O_2$ level at 22 °C was reduced to an extent comparable to wild type in *Mn-SOD* overexpressing plants, and was reduced (but still higher than wild type) in *Cu/Zn-SOD* overexpressing plants. By contrast, overexpressing *Fe-SOD* only slightly decreased the $O_2^{·-}$ accumulation (Fig. 5c). Therefore, the degree of complementation of transgenic plants expressing different types of SOD decreased with increasing $O_2^{·-}$ accumulation (Fig. 5b, c). These results further support the idea that $O_2^{·-}$ accumulation in the leaf blade causes the low temperature–sensitive albino phenotype of *ospus1-1*.

## *sop10* reduces $O_2^{·-}$ accumulation and enhances rice cold tolerance at the seedling stage

Loss of SOP10 function, which affects mitochondrial complex I, suppressed $O_2^{·-}$ accumulation and improved cold tolerance in double mutants with the chloroplast mutant *ospus1-1*. We, therefore, wondered whether *sop10* single mutants could tolerate lower

temperatures than wild-type 9311. To test this, we treated the wild-type 9311 and *sop10* single mutants with three different low temperature conditions (6 °C, 8 °C, and 15 °C). Indeed, the *sop10* plants could survive 8 °C treatment for 2 days, and $O_2^{·-}$ levels in *sop10* mutants were lower than in 9311 under cold treatment and during recovery (Fig. S9a–c). Consistent with the $O_2^{·-}$ accumulation, the mETC complex I capacity in *sop10* mutants was reduced regardless of temperature (Fig. S9d).

We then compared the yield of *sop10* plants with 9311 under normal growth conditions and found that the total yield per plant of *sop10* was similar to that of 9311 (Fig. S9e). These results indicate that the SOP10 regulates redox homeostasis and thereby affects plant growth in response to low temperature conditions, probably by affecting mETC complex I function.

To further test the potential for using SOP10 in breeding for cold-tolerant rice, we created loss-of-function mutants of *sop10* in tropical

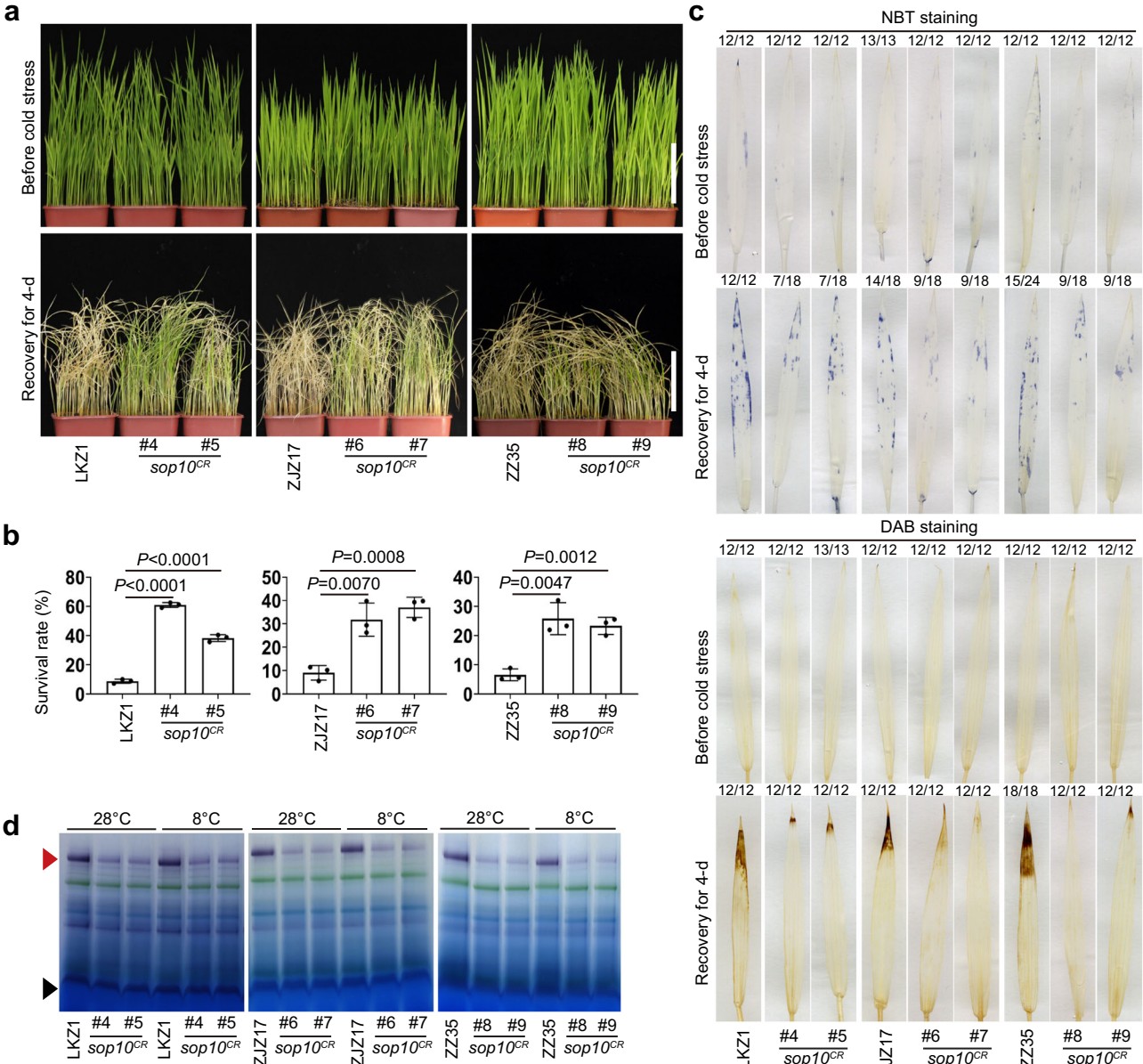

**Fig. 6 | Assessment of cold stress tolerance of *sop10* single mutant plants at seedling stage. a** Seedlings of LKZ1, ZJZ17, ZZ35, and their two corresponding *sop10^CR* mutants were grown at 28 °C for 2 weeks, shifted to 8 °C for 2 days, and allowed to recover at 28 °C for 4 days. Scale bars, 8 cm. **b** Survival rates of LKZ1, ZJZ17, ZZ35, and their *sop10^CR* seedlings after cold treatment in (**a**). Values are means ± S.D. (*n* = 50 individual plants) of three biological replicates. Differences between mutants and wild type grown under the same conditions are determined by unpaired Student's *t*-test (two-tailed). Exact *P*-values are shown. **c** NBT and DAB staining were used to assess the accumulation of $O_2^-$ and $H_2O_2$ in LKZ1, ZJZ17, ZZ35, and their *sop10^CR* mutant seedlings before and after cold treatment in (**a**). The numbers on top of each photo show how often this representative staining pattern occurred, relative to total examined pictures. Scale bars, 1 cm. **d** In-gel assay of NADH oxidase capacity of LKZ1, ZJZ17, ZZ35, and their *sop10^CR* mutants. Dihydrolipoamide dehydrogenase activity was used as a loading control. The red triangle indicates mitochondrial complex I, and the black triangle indicates dihydrolipoamide. Experiments were repeated two times independently with similar results. Source data for (**b**–**d**) are provided as a Source Data file. Experiments were repeated three times with similar results.

or subtropical *indica* rice varieties using CRISPR-Cas9. LKZ1, ZJZ17, and ZZ35 are *indica* rice varieties that are widely planted in southern China and often encounter cold stress at the seedling stage when they are planted in the spring. We generated *sop10* knock-out mutants in LKZ1, ZJZ17, and ZZ35, and evaluated their cold tolerance. As shown in Fig. 6, knocking out *SOP10* in these three *indica* varieties significantly decreased mETC complex I capacity and cellular $O_2^-$ accumulation, improving their survival rates under 8 °C treatment for 2 days at the seedling stage (Fig. 6 and Fig. S10). Collectively, these results show that SOP10 is a negative regulator of cold tolerance and mutations in *SOP10* improved cold tolerance of *indica* rice.

## Discussion

Shifting rice production from transplanting to direct seeding enhances sustainability[57,58]. However, direct-seeded rice usually encounters cold stress at the seedling stage. Improving the cold tolerance of rice varieties at the seedling stage is, therefore, important for developing direct-seeding rice. In this work, we show that knock-out of the *SOP10* enhances cold tolerance at the seedling stage for *indica* rice varieties without a yield trade-off (Fig. 6), providing a target for breeding or engineering direct-seeding *indica* rice. Maize (*Zea mays*) SOP10 shows high amino acid sequence identity (81%) with rice SOP10 and could be used as a potential target for designing cold-tolerant direct-seeding

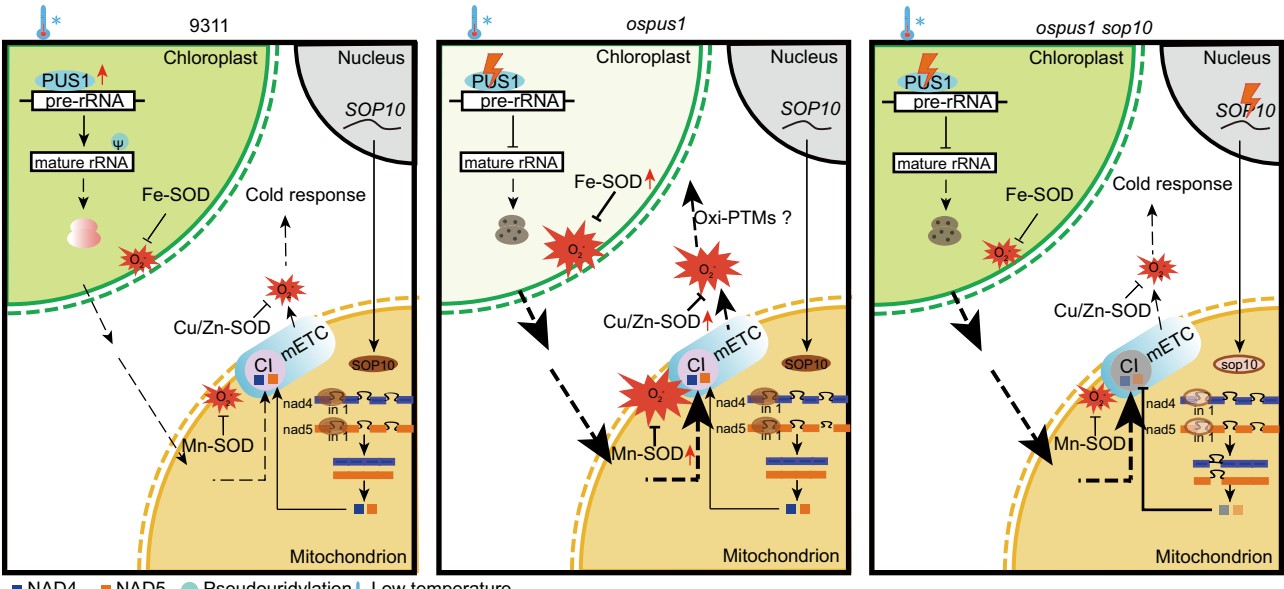

**Fig. 7 | A proposed model of albino formation mediated by *ospus1* under low temperature.** (Left) In wild-type 9311, PUS1 is upregulated and catalyzes the pseudouridylation of pre-rRNAs under cold conditions, enabling normal chloroplast ribosome assembly and translation, thus maintaining homeostasis in chloroplast metabolism. (Middle) A deficiency in chloroplast pre-rRNA pseudouridylation in *ospus1* mutants results in aberrant chloroplast ribosomes and defects in plastid translation, which then leads to an imbalance of chloroplast metabolism. The disruption of chloroplast homeostasis generated a chemical/signal that was transmitted into mitochondrion and induced $O_2^{\cdot-}$ overproduction from the mitochondrial ETC (mETC) complex I. The mitochondrial $O_2^{\cdot-}$ then caused the albino phenotype in *ospus1* under low temperatures, by oxidizing target proteins (Oxi-PTMs), which remain to be identified. (Right) Mutations in a mitochondrial PPR protein (SOP10) that directly binds to *nad4* and *nad5* transcripts and regulates splicing of their first exons, decreases mETC complex I capacity and $O_2^{\cdot-}$ accumulation, and suppresses the albino phenotype of *ospus1*, enhances rice cold tolerance in wild-type 9311.

maize. In this study, by screening for *ospus1* suppressors, we identified a cold-tolerance gene, showing that the *ospus1* mutant could be used to screen for additional cold-tolerant genetic resources.

A dysfunction in chloroplasts (*ospus1*) results in ROS accumulation and an albino phenotype under low temperatures[12]. Mutations in the mitochondrial protein SOP10 reduced the levels of the NAD4 and NAD5 proteins, which are components of complex I, resulting in less functional complex I and less $O_2^{\cdot-}$ production, and suppressed the phenotype caused by *ospus1* (Fig. 7). Furthermore, overexpression of the gene encoding a mitochondrial Mn-SOD rescued the albino phenotype in *ospus1* (Fig. 5). These findings indicate that mitochondrion-generated $O_2^{\cdot-}$ plays a central role in the *ospus1-1* albino phenotype. This is very similar to a previous report of the *Arabidopsis mod1* mutant, in which a deficiency in fatty acid biosynthesis in chloroplasts resulted in mitochondrial ROS accumulation and cell death, which were suppressed by a mutation in a mitochondrial protein, suggesting chloroplast-to-mitochondria communication in *mod1*-mediated programmed cell death[59]. Further study revealed that malate-mediated chloroplast-to-mitochondria communication regulated programmed cell death: the deficiency in *mod1* chloroplasts produced elevated malate in chloroplasts. The malate was then transported into mitochondria and oxidized to generate NADH, which serves as the electron donor for the mtETC to generate ROS, triggering programmed cell death. Mutations in any component of the malate shuttle pathway between chloroplasts and mitochondria block the formation of mitochondrial ROS and suppress *mod1* phenotypes[60]. Whether the malate shuttle participates in the *ospus1* albino phenotype, or another malate-independent chemical is emitted from chloroplasts and transported into mitochondria, needs further investigation (Fig. 7). If the communication between chloroplasts and mitochondria participates in cold responses, any mutation in the communication pathway would be expected to enhance cold tolerance.

Further, how mitochondrial $O_2^{\cdot-}$ causes abnormal chloroplast development and albino phenotypes under low temperatures is another important question. As a signal molecule, changes in $O_2^{\cdot-}$ levels can alter the structure and function of multiple proteins and therefore impact many different signal transduction pathways. To understand how the $O_2^{\cdot-}$ causes the albino phenotype in *ospus1-1* and regulates cold responses, further identification of the downstream oxidative posttranslational modified proteins is required.

The NBT staining for $O_2^{\cdot-}$ appears in the leaf blade, but the DAB staining for $H_2O_2$ only appears at the leaf tip in *ospus1-1* mutants at 22 °C. The albino phenotype occurs in the whole leaf blade in *ospus1-1*, which is consistent with the $O_2^{\cdot-}$ accumulation pattern (Fig. 4a, b), indicating that the albino phenotype may be caused by the accumulated $O_2^{\cdot-}$. Since rice leaves develop from base to tip, cells of the leaf tip are developmentally the oldest in the leaf, and the albino phenotype in *ospus1-1* leads to earlier senescence of the leaf tip than the wild-type 9311 at 22 °C (Fig. 4b). Therefore, the accumulation of $H_2O_2$ in the leaf tip in *ospus1-1* may be senescence induced. These data suggested that the $O_2^{\cdot-}$ accumulation caused the albino phenotype in *ospus1-1*.

This is supported by the overexpression of SOD isoforms, which alleviated the $O_2^{\cdot-}$ accumulation in *ospus1* under cold stress and rescued the albino phenotype to different extents. Overexpression of mitochondrial Mn-SOD suppressed the low temperature–induced $O_2^{\cdot-}$ accumulation in *ospus1-1* to the greatest extent, removing the accumulated $O_2^{\cdot-}$ in the leaf blade with only light NBT staining in the leaf tip. Consistent with its suppression of $O_2^{\cdot-}$ accumulation, Mn-SOD rescued the *ospus1* albino phenotype to a greater degree than the cytosolic Cu/Zn-SOD. By contrast, overexpression of *Fe*-SOD slightly decreased $O_2^{\cdot-}$ accumulation in the leaf blade, rescuing the albino phenotype to a much lesser degree than the other two SODs, although its transcript level was about 30-fold higher than non-transgenic plants (Fig. 5). These data indicate that the albino phenotype was stronger in plants with higher cellular $O_2^{\cdot-}$ levels under cold stress. The overexpression of mitochondrial SOD suppressed $O_2^{\cdot-}$ accumulation and the albino phenotype the best, and the *sop10* mutant suppressed accumulation of $O_2^{\cdot-}$ via reducing the complex I activity of mETC, which produces $O_2^{\cdot-}$

as the primary ROS. Together, the $O_2^{\cdot-}$ in the mitochondria leads to the albino phenotype under cold stress and regulates the cold stress response.

Most PPR proteins implicated in intron splicing in the mitochondria reported to date are from *Arabidopsis* and maize. Many proteins involved in the splicing of mitochondrial introns *nad4i461g2* and *nad5i230g2* have also been reported. For example, DEK35[32], DEK43[30], DEK55[33], Emp8[34], Emp24, Emp25[35], Emp602[36], PPR18[37], PPR-SMR1[38] in maize, and OZ2[61] in *Arabidopsis*, are involved in the splicing of *nad4* intron 1 (*nad4i461g2*). mCSF1, PPR101 and PPR231 in maize are required for the splicing of *nad5* intron 1 (*nad5i230g2*) and intron 2 (*nad5i1455g2*)[62]. Arabidopsis MISF68[31], ABO6[63] affect both *nad4i461g2* and *nad5i230g2* splicing. Mutants of these PPR genes lead to complete abolishment of intact complex I and result in severe seed developmental defects, highlighting the importance of complex I for seed development. By contrast, only a few proteins in rice were identified to function in mitochondrial intron splicing, such as OsPPR939 for *nad5* intron 1, 2, 3[40] and RL1 for *nad4* intron 1[41]. Here, we identified SOP10 and showed it participates in splicing of *nad4* and *nad5* intron1 and RNA editing of *nad2* and *nad6*. Mutants of *SOP10* showed enhanced cold tolerance with no developmental defect in rice (Fig. 6 and Figs. S9, 10), probably because it only moderately affects the activity of functional complex I. From this perspective, SOP10 has a weaker effect on complex I assembly, suggesting that we can expect to obtain enhanced tolerance to cold or other stress similar to SOP10 by screening for other weak alleles of complex I assembly factors.

Together, our observations demonstrate that the cold stress-triggered overproduction of $O_2^{\cdot-}$ from mitochondria provides a target for manipulating ROS. Importantly, we identified a ROS regulator, the mutation of which reduces ROS production; these mutations can be applied to improve cold tolerance in rice.

# Methods

## Plant materials and growth conditions
Rice seedlings (*Oryza sativa* ssp. *indica* cultivar 9311) were grown in growth chambers (14 h light/10 h dark cycle with 70% humidity) at the indicated temperatures. For cold treatment, the chamber temperature was set at 22 °C after the seeds germinated. To evaluate yield performance after cold stress, we transplanted the seedlings from the chambers to the paddy field (Beijing) after cold treatment until the grains were ripe.

The *sop10^{CR}* single mutants in the 9311 background were generated by crossing *ospus1-1 sop10^{CR}* with 9311 and genotyping to select homozygous *sop10^{CR}*/9311 genotype.

## Screening for suppressors of *ospus1-1*
To identify *suppressor of ospus1-1* (*sop*) mutants, 10,000 seeds of *ospus1-1* were mutagenized with ethylmethane sulfonate (EMS, 0.75%, v/v) for 16 h. M₁ and M₂ plants with the green leaves were selected in the growth chamber at 22 °C.

## Mapping-by-sequencing
For the cloning of *SOP*, *ospus1-1 sop10-1* was backcrossed with *ospus1-1* to generate F₂ populations. Individual plants showing green leaves at 22 °C were selected from F₂ populations and pooled for DNA sequencing. For reference, a pool of *ospus1-1* plants was also collected. DNA from *ospus1-1* and *sop10-1* were used for whole-genome sequencing using the Illumina HiSeq2500 platform.

The sequence reads were quality controlled with Trimmomatic and were aligned to the reference Rice Genome (IRGSP-1.0) using BWA-MEM with default parameters. Mapped reads were then sorted with samtools and duplicates were marked with PicardTools (https://broadinstitute.github.io/picard). SNP calling was based on alignment results using samtools/bcftools and further filtered with mapping quality ≥30. The SNP index (proportion of reads harboring a SNP that differ from the reference sequence) was calculated and plotted across the 12 rice chromosomes to identify candidate regions for each pool.

## Plasmid construction and plant transformation
For genetic complementation, a fragment containing the full-length genomic DNA of *SOP10* with its native promoter region, starting 2500 bp upstream of the start codon and ending 1500 bp downstream of the stop codon was amplified and cloned into the binary vector pCAMBIA1300.

The CRISPR-Cas9 system was used to generate knockout mutants of *SOP10* in the *ospus1-1* background[64]. For subcellular localization, the fragment containing the full-length coding sequences (CDS) of *SOP10* and *sop10-1* were amplified and cloned into pRTVnGFP[65] to generate *p35S:OsSOP10-GFP* and *p35S:OsSOP10-1-GFP*.

For overexpression of SOD genes, CDS of *Mn-SOD* (LOC_Os05g25850), *Fe-SOD* (LOC_Os06g05110) and *Cu/Zn-SOD* (LOC_Os07g46990) were amplified and cloned into the binary vector XF426m with the maize *Ubiquitin* promoter. The final binary vectors were introduced into the *Agrobacterium tumefaciens* strain EHA105 and transformed into rice calli. All primers and oligonucleotides used are listed in Supplementary Table 1.

## Subcellular localization
To study the subcellular localization of OsSOP10 and OsSOP10-1, protoplasts were prepared from 2-week-old rice seedlings grown at 28 °C[66]. OMP25-RFP was used as the mitochondrial marker protein, as previously reported[27]. Fluorescent signals were examined with a LSM980 confocal microscope (Carl Zeiss) with emission/detection settings at 488/585–615 nm to detect GFP and 558/570–623 nm to detect RFP.

## RNA isolation and expression analysis
For RT-qPCR analysis, leaves of rice grown at 22 °C or 28 °C were collected and immediately flash-frozen in liquid nitrogen. Total RNA was isolated using the TRIzol Kit (Invitrogen, Cat #15596018) according to the user's manual. A total of 2 µg RNA was used for cDNA synthesis using the All-In-One 5× RT MasterMix with gDNA Eraser (ABM, Cat #G592). The synthesized cDNA was diluted 10-fold and a 2.0 µL aliquot was subjected to RT-qPCR using SsoFast EvaGreen Supermix (Bio-Rad). *OsACTIN1* was used as a reference for RT-qPCR. All the primers used for RT-qPCR are listed in Supplementary Table 1.

## Northern blot
Northern blotting was performed as previously described in ref. 67. A total of 5 µg of total RNA was separated on a 1.2% (w/v) agarose/formaldehyde gel, then transferred to a Hybond N⁺ membrane (GE Healthcare) by capillary elution. Hybridization was performed overnight at 45 °C in Church buffer. The blots were washed and exposed to a storage phosphor screen (GE Healthcare), then signals were detected with a Typhoon TRIO scanner (GE Healthcare).

## Assay for mitochondrial complex I capacity
NADH oxidase activity of mitochondrion complex I was measured as previously described with some modifications[60]. Rice seedlings were grown for two weeks and the leaves were used for extraction of crude proteins. In brief, 15 µg protein from each sample was separated by 4% to 20% gradient Blue-Native polyacrylamide gel electrophoresis (BN-PAGE) according to the manufacturer's instructions. After electrophoresis, the BN-PAGE gels were stained with Coomassie Brilliant Blue R-250 followed by incubation in the staining buffer (50 mM MOPS-KOH, pH 7.6, 1 mM nitro blue tetrazolium and 0.2 mM NADH). The reaction was stopped when the dark blue stain was strong enough and the gel was washed in deionized water for 16 h with gentle shaking at 50 rpm.

## Histochemical staining for H₂O₂ and O₂·⁻

$H_2O_2$ and $O_2^{\cdot-}$ in the leaf were detected using 3, 3′-diaminobenzidine (DAB) and nitro blue tetrazolium (NBT), respectively, as previously described in ref. [60]. In brief, 2-week-old seedlings were immersed in freshly prepared 1 mg/mL DAB staining solution (pH 3.8) or 50 mM sodium phosphate (pH 7.0) containing 0.05% NBT and 10 mM $NaN_3$ and infiltrated under vacuum at 0.8 psi for 30 min. Then the seedlings were incubated at 25 °C in the dark for 6 h with gentle shaking at 100 rpm. After infiltration, the seedlings were bleached (acetic acid: glycerol: ethanol = 1:1:3, v/v/v) at 65 °C for 2 h, and stored in 95% (v/v) ethanol until scanning.

## SOD in-gel activity assay

SOD activity was detected by an in-gel native PAGE assay[68]. Briefly 0.2 g samples were homogenized in protein extraction buffer (0.1 mM EDTA, 0.1% (w/v) Triton-X100, 1 mM PMSF, and 1% [w/v] PVP) on ice for total protein extraction. Then, 20 μg of protein was loaded on the 10% Native PAGE and electrophoresed at a constant voltage of 100 V for 1.5 h. The gel was incubated in 10 mL of Riboflavin-NBT solution (0.26 mM Riboflavin, 0.3 mM NBT) at room temperature for 20 min in the dark. The Riboflavin-NBT solution was removed and the gel was incubated in 20 mL of 0.1% (v/v) TEMED at room temperature for another 20 min in the dark. Superoxide was induced under light before scanning.

## RIP RT-PCR assay

The rice calli harboring $Pro_{Ubq}$:OsSOP10-3×Flag were cultured at 28 °C and used for the RIP assay[69]. In brief, 1 g of calli was ground into fine powder with liquid nitrogen and crosslinked at 800 mJ/$cm^2$ in a Hoefer UVC 500 ML Ultraviolet Crosslinker (GE). Next, 1 mL of extraction buffer (1×PBS, 1% CA630, 1 mM PMSF, 1 mM DTT, RNase inhibitor (NEB, S1402S) with 1× protease inhibitor cocktail (Sigma-Aldrich, P9599) was added to each sample and incubated for 15 min at 4 °C. Then the supernatant was collected by centrifuging at 12,000 g for 30 min at 4 °C, and used for immunoprecipitation with ANTI-FLAG M2 Magnetic Beads (M8823, Sigma) for 3 h at 4 °C. Finally, the immunoprecipitated RNA was extracted and reverse transcribed with HiScript II 1st Strand cDNA Synthesis Kit (Vazyme, R212-01). The cDNAs were used as templates for RT-PCR with gene-specific primers (Supplementary Table 1) to detect the target transcripts.

## STS-PCR-seq assay

The above-ground parts from wild-type and sop10-2 seedlings at 12 DAP were used to isolate total RNA. Each RNA sample was extracted from six independent seedlings. Two biological replicates were used to analyze editing by STS-PCR-seq. To this end, the total RNA was reverse transcribed using random primers to generate cDNA. The 34 protein-coding mitochondrial transcripts were amplified from the cDNA, and the RT-PCR products were mixed in equimolar ratios. High-throughput sequencing and data analysis were performed according to Wang et al.[70]. The threshold for a decreased editing efficiency was defined as [T/(T + C)% in the wild type - T/(T + C)% in the mutant] ≥ 20%. The primers used for RT-PCR to amplify the mitochondrial genes are in Supplementary Table 2.

## Statistical analysis

In this study, all data were analyzed by two-tailed Student's t-test or analysis of variance (ANOVA), as noted in the text. P values less than 0.05 were considered statistically significant. Details and numbers of biological replicates are described in the respective figure legends.

## Data availability

Data supporting the findings of this work are available with in the paper and its Supplementary Information files. The DNA-resequencing dataset has been deposited in the Genome Sequence Archive (GSA, Genomics, Proteomics & Bioinformatics 2021) in National Genomics Data Center (Nucleic Acids Res 2022), China National Center for Bioinformation/ Beijing Institute of Genomics, Chinese Academy of Sciences under accession code CRA011784. The STS-PCR-seq data are available under accession number CRA012645. Source data are provided with this paper.

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

## Acknowledgements

This work was supported by grants from the Chinese Academy of Sciences (Strategic Priority Research Program XDA24010302-2 to X.S.), the National Natural Science Foundation of China (No.32372050 to L.L., No.32000446 to L.L.), the National Key Research and Development Program of China (2020YFA0509900 to X.C.), the Chinese Academy of Sciences (Strategic Priority Research Program XDB27030201 to X.C.).

## Author contributions

X.C. and L.L. conceived and supervised the project, analyzed the data, and wrote the paper with the inputs from all authors; X.Z. and L.L. performed experiments; Z.W. performed some of the experiments and helped to write the paper. J.G. and X.S. created the EMS mutant population. C.Y. and C.X. performed the bioinformatics analysis. Y.W., X.Q., X.D., C.C., and B.T. helped to analyze the data and write the paper.

## Competing interests

The authors declare no competing interests.
