## [Peer Review File · Nature Communications]

A mitochondrial pentatricopeptide repeat protein enhances cold tolerance by modulating mitochondrial superoxide in riceREVIEWER COMMENTS

Reviewer #1 (Remarks to the Author):

The studies reported in this manuscript build on previous work by these authors on the mitochondria-localised pentatricopeptide repeat (PPR) protein called pseudouridine synthase, OSPUS1-1, which binds to transcripts encoding the mitochondrial complex I subunits NAD4 and NAD5. The mutant is albino under low temperatures because of low complex I activity and altered chloroplast ribosome biogenesis. In the present study, the authors report the isolation of a suppressor of *ospus1-1*, which partially rescues the low temperature-sensitive leaf albino phenotype of the *ospus1-1* mutants. The authors had previously reported that the *ospus1-1* mutants show enhanced ROS accumulation. The authors have opted for a very traditional view of the roles of ROS in plants. Current concepts consider that all ROS are potentially signalling molecules that fulfil essential roles in plant, growth and defence. Within this context, the notion that there are safe and harmful levels of ROS is inaccurate. Moreover, ROS accumulation does not only occur under stress conditions. Moreover, current data suggest that catalase (CAT) only acts to remove hydrogen peroxide in peroxisomes and that the broad network of antioxidant defences accomplish this task in other organelles. While peroxidases are relevant to this task, peroxidases can also produce hydrogen peroxide, and so this is not the most relevant enzyme to measure. Crucial to data interpretation within this context is that superoxide dismutases (SOD) merely convert one weakly reactive ROS form (superoxide) to another more reactive ROS form (hydrogen peroxide). The premise, as stated in lines 217-218 that it may be possible to decrease ROS levels by overexpressing SOD in *ospus1-1*, is poorly informed. I wonder therefore if data interpretation has been biased by the above conceptual ideas. I have some concerns about the description of the data and data interpretation as outlined below:

The authors expressed Mn-SOD, Fe-SOD, and Cu/Zn-SOD in the mutants and presented evidence that all four SOD isoforms were significantly higher in the transformed *ospus1-1* mutants than the wild type and other lines. I find the data in Figure 4 rather difficult to understand. Crucially, the patterns of NBT and DAB staining in the leaves. While the NBT staining appears in the leaf blade the DAB stain only accumulates at the leaf tip. These details are not discussed in the text. Since rice is a monocot species, the leaf cells develop from base to tip. Hence, hydrogen peroxide only accumulates in the oldest cells of the leaf tip. This is evident in the leaves from the *ospus1-1* mutant but it is not discussed. The findings are interesting finding because they suggest that OsPUS1 regulates only the senescence -induced accumulation of hydrogen peroxide.

Similarly, the data shown in Figure 5 concerning how the overexpression of SOD genes suppresses ROS accumulation is equally intriguing but not discussed appropriately. Unlike the NBT data shown in Figure 4, the data in Figure 5, show that NBT staining occurs mainly in or around the leaf tip, again suggesting that the overexpression of MnSOD and CuZn SOD but not FeSOD protects against the senescence-induced accumulation of hydrogen peroxide at the leaf tip. The authors should also discuss why the pattern of NBT staining in the wild type and mutants is different in Figure 4 and Figure 5.

The data in Figure 5B clearly show that the albino leaf phenotype observed in the *ospus1-1* mutants is prevented by overexpression of overexpression of any of the SOD forms. In contrast, I find the data shown in Figure 6 regarding decreased ROS accumulation in the SOP10 knockouts in three indica varieties less convincing particularly because the pattern of NBT staining is not consistently changed between the lines.

Based on the above comments I am not convinced that changes in ROS accumulation link provide any evidence for their contribution to mitochondrial and chloroplast communication

as discussed by the authors. The data demonstrate that the lack of the mitochondrial PPR protein SOP10 rescues the albino phenotype. However, the conclusion that this involves chloroplast-to-mitochondrion (CTM) communication that regulates cellular ROS levels at low temperatures is far too general to be accurate. As presented, the discussion section is weak and uninformative. It does not provide a critical evaluation of the data or the mechanisms involved.

and that the overexpression of SOD protects against the senescence -induced accumulation of hydrogen peroxide.

The authors present data showing that the levels of glutathione and CAT and POD activities were also significantly higher in the transformed *ospus1-1* plants than other lines. The data for enzyme activity are expressed as units per mg fresh weight but the authors do not provide the definition of a Unit in terms of activity. This information should be provided. Similarly, the authors only provide data on reduced glutathione (GSH). It would be more informative to provide data on both GSH and glutathione disulphide. This would allow a more accurate understanding of changes in the total glutathione pool and its oxidation state. The data do not shed any new light on the mechanisms involved in mitochondria to nucleus communication. The authors should be careful not to overinterpret their data on this point.

Reviewer #2 (Remarks to the Author):

Comments on „Chloroplast-to-mitochondrion communication regulates reactive oxygen species production and cold responses in rice“ by Zu et al., submitted to Nat. Comm.

The authors build upon their earlier work, published in *New Phytologist* last year, characterizing a cold-sensitive, albino rice mutant affected in a chloroplast pseudouridine synthase OsPUS1, which is necessary for cp rRNA maturation at lowered temperatures of 22° vs. 28°C. The cold-sensitive PUS1 phenotype is associated with increased ROS (reactive oxygen species) accumulation. Screening for genetic suppressors in an EMS-mutagenized population they identify SOP10, encoding a mitochondrial pentatricopeptide repeat (PPR) protein. SOP10 suppresses the cold-sensitive phenotype but does not reconstitute rRNA processing in chloroplasts. Instead, SOP10 turns out to affect splicing of the respective first introns in mitochondrial *nad4* and *nad5* transcripts resulting in reduced complex I activities in the SOP10 mutants. Expressing different SODs (superoxide dismutases) can compensate the cold-sensitive phenotype and independently mutating SOP10 in three different and widely used rice indica varieties likewise lowers cold-sensitivity and ROS levels.

The manuscript summarizes solid experimental work and is clearly written. However, in my opinion the paper suffers from overstatements assuming “regulation” and intracellular “communication”, already starting with the paper’s title. As in similar examples in the recent scientific literature such terms often appear overused to “sell” the respective stories (see e.g., the last two sentences of the abstract). Isn’t it simply that mutation of SOP10 causes a moderate defect of mitochondrial complex 1, which results in lower ROS levels, and which is able to ameliorate the PUS1 phenotype? In that context, also note the sentence starting in line 90 speaking of “a mutation in a mitochondrial protein”, which is simply not the case here, where we just find a moderate reduction in mature mRNA levels for *nad4* and *nad5*. Along

these lines, a display like figure 7 postulating a messenger molecule and labeling it with a question mark is not helpful.

The above comments do not mean to suggest that the results are not fine and interesting, but I feel that the claims go simply too far and are not quite of the wide general interest that would seem fully adequate for the readership of Nature Communications.

Some further comments:

1. How can a G1475A mutation result in a G492N amino acid change (line 112f. and figure 1)? The change of GGC to GAC would instead cause a G-to-D codon conversion, wouldn't it?
2. It should be mentioned that the splicing defect affects group II introns (BTW: introns would be named nad4i461g2 and nad5i230g2, respectively, following a general nomenclature proposal for plant organellar introns).
3. The authors offer a rather arbitrary selection of references on PPR proteins (their references 34 to 38, cited in line 148). Here, it would be especially interesting to let readers know about some of the other protein factors that already have previously been shown to affect splicing of exactly these two mitochondrial introns in model systems like Arabidopsis (ABO6, MISF68, OZ2), maize (DEK35, DEK55, EMP8, EMP24, EMP25, EMP602, PPR18, PPR-SMR1) or even rice (FSE5, PPR939, RL1)
4. The above is quite important in this context, and the authors may find it worth discussing whether they would expect to find phenotypes similar to the one of SOP10 for "mild" mutant alleles of those factors, too.
5. Strangely, the authors mention PLS-type variants of PPR proteins (lines 133ff.) but then rely on a rather old/outdated PPR prediction algorithm, not taking into account modern PPR HMMer profiles. Here, the alternative modern approach (see <https://ppr.plantenergy.uwa.edu.au/>) comes to quite different conclusions on the makeup of SOP10, not suggesting 13 P-type PPRs but rather 18 PLS-type (including SS-type) PPRs and, most notably, also terminal E1 and E2 motifs that are otherwise frequently signatures of organelle RNA editing factors.
6. In the above context it would be interesting to learn whether the authors have used the known PPR-RNA recognition code to make prognoses about where exactly SOP10 may bind to the nad4 and nad5 mRNAs.

Reviewer #3 (Remarks to the Author):

The manuscript by Zu et al. titled "Chloroplast-to-mitochondrion communication regulates reactive oxygen species production and cold responses in rice" describes a genetic screen to identify a suppressor of the low-temperature albino phenotype due to a mutation in OsPUS1, a plastid localized pseudouridine synthase coding gene involved in chloroplast biogenesis. The *ospus1* mutant has elevated ROS levels at 22°C compared to 28°C, correlating with the albino phenotype. The identified suppressor of *ospus1-1* gene SOP10 codes for a pentatricopeptide repeat (PPR) protein regulating splicing of

mitochondrial genes *NAD4* and *NAD5* , leading to deficiencies in complex I and decreased ROS generation, reversing the albino phenotype at 22°C. Moreover, overexpression of SOD genes localizing to different cellular departments also reversed the albino phenotype of the *ospus1* mutant, suggesting that ROS accumulation was responsible for the phenotype. Lastly, the authors show that CRISPR/Cas9 generated single mutants in *SOP10* improved the cold tolerance potential of three cold sensitive rice cultivars without a yield penalty.

This manuscript is a follow-up of their 2022 paper on identifying *OsPUS1* as a pseudouridine synthase required for proper chloroplast function at mild chilling temperatures. It presents an interesting story showing evidence for chloroplast-to-mitochondrion communication, which appears to be required under abiotic stress conditions to maintain ROS homeostasis. The manuscript is well written, clear and concise, and the interpretations fit the results. While it was previously shown that PPR proteins regulate RNA metabolism of plants organelles required for proper organelle biosynthesis under abiotic stress conditions, this manuscript provides evidence suggesting that chloroplast-to-mitochondrion communication regulates cold responses via ROS homeostasis even at low chilling temperatures.

To clarify a few questions I have, please address the following points:

1. Lines 164-166: The statement that “[...] results suggest that *SOP10* directly binds to *nad4* and *nad5* transcripts and participates in the splicing of the first intron of their mRNAs in a temperature-independent manner” is correct based on Fig. 2 and Fig. S4. However, it seems that only primers interrogating intron 1 were used in Fig. S4. Were the other introns investigated as well? If so, please state; if not, please qualify your statements indicating that possibly other introns are retained as well in the *sop10* mutant. There is no size marker in Fig. 2c, but it looks like the entire transcript might not be spliced, therefore, please add size markers to Fig. 2 to determine the size of the unspliced transcript. Also, *nad5* has two splice products in 9311 and the larger one is reduced in the *sop10* mutant. Please address and explain the significance of the two splice forms and why abundance of the larger one, possibly containing unspliced introns, is reduced.
2. Fig. 6: legend states “allowed to recover...for 4 days” while figure labeling says, “recover for 2-d”. Also note that “recovery” would be better than “recover” in the figure labeling.

Reviewer #1 (Remarks to the Author):

The studies reported in this manuscript build on previous work by these authors on the mitochondria-localised pentatricopeptide repeat (PPR) protein called pseudouridine synthase, *OSPUS1-1*, which binds to transcripts encoding the mitochondrial complex I subunits NAD4 and NAD5. The mutant is albino under low temperatures because of low complex I activity and altered chloroplast ribosome biogenesis. In the present study, the authors report the isolation of a suppressor of *ospus1-1*, which partially rescues the low temperature-sensitive leaf albino phenotype of the *ospus1-1* mutants. The authors had previously reported that the *ospus1-1* mutants show enhanced ROS accumulation.

1. The authors have opted for a very traditional view of the roles of ROS in plants. Current concepts consider that all ROS are potentially signaling molecules that fulfil essential roles in plant, growth and defense. Within this context, the notion that there are safe and harmful levels of ROS is inaccurate. Moreover, ROS accumulation does not only occur under stress conditions.

Response: We thank the reviewer for pointing out this issue. ROS play important roles in plants, controlling processes such as growth, development and especially responses to biotic and abiotic stress stimuli. Truly, ROS accumulate and act as signals in both plant growth (in stem cell maintaining) and development (in cell differentiation), and stress responses. We updated our description about ROS in developmental regulation and added new references accordingly (Lines 55–65 in the revised manuscript).

2. Moreover, current data suggest that catalase (CAT) only acts to remove hydrogen peroxide in peroxisomes and that the broad network of antioxidant defenses accomplish this task in other organelles. While peroxidases are relevant to this task, peroxidases can also produce hydrogen peroxide, and so this is not the most relevant enzyme to measure.

Response: We thank the reviewer for the constructive comments. In the previous version, we did not explain well the difference between NBT and DAB staining. After re-analyzing our original data and repeating this experiment, we confirmed the NBT and DAB staining patterns in *ospus1*. As the reviewer suggested, we indeed found that the NBT staining signal appears in the whole leaf blade, which is consistent with the albino phenotype of *ospus1*. By contrast, the DAB staining signal only accumulates at the leaf tip. The albino phenotype was related to the accumulated $O_2^{\cdot-}$ in the leaf blade. This difference suggest that the $O_2^{\cdot-}$ is more relevant/important to the albino phenotype. We summarized the relationship between ROS staining and the albino phenotype as a schematic drawing, which was added in the revised manuscript (Fig. 4b). We believe that assessing the hydrogen peroxide scavenging system may not be helpful for fully understanding the albino phenotype. We, therefore, removed the related data (original Fig. 4c) in the revised manuscript.

3. Crucial to data interpretation within this context is that superoxide dismutases (SOD) merely convert one weakly reactive ROS form (superoxide) to another more reactive ROS form (hydrogen peroxide).

Response: We thank the reviewer for the suggestion. As we know, hydrogen peroxide (H_2O_2) is more stable than superoxide ($O_2^{\cdot-}$), it can diffuse or be transported within a cell or between cells. We detected the H_2O_2 levels in leaves of

ospus1-1 overexpressing different SOD isoforms. We found that the levels of H₂O₂ decreased in SOD OE/*ospus1-1* transgenic rice as shown by DAB staining. Therefore, overexpression of SOD in *ospus1-1* decreased ROS levels including both O₂^{•-} and H₂O₂. Our data was consistent with a previous concept that overexpression of SOD does not simply lead to increased H₂O₂, because the presence of very efficient enzymatic pathways designed to keep H₂O₂ levels low, including catalase, ascorbate peroxidase (APX), and glutathione peroxidases (GPX) (MacMillan-Crow and Crow, 2011). We added the new data into the revised Fig.5c. in the revised manuscript.

4. The premise, as stated in lines 217-218 that it may be possible to decrease ROS levels by overexpressing SOD in *ospus1-1*, is poorly informed. I wonder therefore if data interpretation has been biased by the above conceptual ideas.

Response: As aforementioned, the ROS staining data indicated a close link between O₂^{•-} accumulation and the albino phenotype in *ospus1-1* (Fig. 4a, b in revised manuscript). To test whether the low-temperature sensitive albino phenotype in *ospus1-1* is caused by accumulated O₂^{•-}, we overexpressed SOD which scavenges O₂^{•-} in *ospus1-1* (see Lines 228-231 in revised manuscript).

5. I have some concerns about the description of the data and data interpretation as outlined below:

The authors expressed Mn-SOD, Fe-SOD, and Cu/Zn-SOD in the mutants and presented evidence that all four SOD isoforms were significantly higher in the transformed *ospus1-1* mutants than the wild type and other lines. I find the data in Figure 4 rather difficult to understand. Crucially, the patterns of NBT and DAB staining in the leaves. While the NBT staining appears in the leaf blade the DAB stain only accumulates at the leaf tip. These details are not discussed in the text. Since rice is a monocot species, the leaf cells develop from base to tip. Hence, hydrogen peroxide only accumulates in the oldest cells of the leaf tip. This is evident in the leaves from the *ospus1-1* mutant but it is not discussed. The findings are interesting finding because they suggest that OsPUS1 regulates only the senescence -induced accumulation of hydrogen peroxide. (The reviewer suggested to describe and discuss the data of NBT and DAB staining patterns in Fig. 4)

Response: We thank the reviewer for the constructive suggestions. We first confirmed the NBT and DAB staining patterns in 9311 and *ospus1-1* plants growing at 22°C, by repeating the ROS staining experiments in Fig. 4. The NBT staining signal reproducibly appears in the leaf blade and the DAB staining reproducibly only accumulates at the leaf tip in *ospus1-1* mutants at 22°C. Combining those observations with the observation that the albino phenotype occurs in the leaf blade but not only the leaf tip, we hypothesized that the albino phenotype is caused by O₂^{•-} accumulation. By contrast, the H₂O₂ accumulation at the leaf tip may be related to senescence, since the cells at leaf tip are developmentally oldest. We described the data in the Result part (Lines 204–212 in the revised manuscript) and interpreted the data in Discussion (Lines 321–329 in the revised manuscript)

6. Similarly, the data shown in Figure 5 concerning how the overexpression of SOD genes suppresses ROS accumulation is equally intriguing but not discussed appropriately. Unlike the NBT data shown in Figure 4, the data in Figure 5, show that

NBT staining occurs mainly in or around the leaf tip, again suggesting that the overexpression of MnSOD and CuZn SOD but not FeSOD protects against the senescence-induced accumulation of hydrogen peroxide at the leaf tip. (The reviewer suggested to discuss the NBT staining data in Fig. 5)

Response: We thank the reviewer for the suggestions. As shown in revised Fig. 5c, overexpression of SOD genes alleviated the $O_2^{\cdot-}$ accumulation in *ospus1* to different extents. Mitochondrial Mn-SOD overexpression suppressed the $O_2^{\cdot-}$ accumulation in the leaf blade and the albino phenotype best, indicating that the albino phenotype is caused mainly by the mitochondrial superoxide. We also examined the H_2O_2 accumulation by DAB staining and found again that the H_2O_2 accumulation occurs at the leaf tip. We discussed the ROS staining data in SOD overexpression plants in the revised manuscript (Lines 330–345 in the revised manuscript)

7. The authors should also discuss why the pattern of NBT staining in the wild type and mutants is different in Figure 4 and Figure 5. The data in Figure 5B clearly show that the albino leaf phenotype observed in the *ospus1-1* mutants is prevented by overexpression of overexpression of any of the SOD forms.

Response: We thank the reviewer for the comments. We repeated the experiment in Fig. 5c, NBT and DAB staining experiments in wild type and *ospus1-1* at 22°C. We examined at least 12 leaves for each condition, and used the representative pattern for wild type and mutants. We found that the representative pattern of NBT staining in wild type and the mutant is consistent in Fig. 4 and Fig. 5. We replaced the pictures in original Fig. 5c with new representative ones (see revised Fig. 5c).

8. In contrast, I find the data shown in Figure 6 regarding decreased ROS accumulation in the SOP10 knockouts in three *indica* varieties less convincing particularly because the pattern of NBT staining is not consistently changed between the lines.

Response: We also repeated the experiment in Fig. 6c with additional DAB staining in SOP10 knockouts in three *indica* varieties under cold treatment (revised Fig. 6c). We analyzed the staining data statistically and found that the NBT staining signal accumulated in the entire leaf blade in all the wild-type plants of three *indica* varieties at 8°C. In the SOP10 knockouts, the two examined lines of each variety showed statistically consistent NBT staining patterns, with staining signal appearing mainly in or around the leaf tip (revised Fig. 6c). We replaced the NBT staining pictures in Fig. 6c with their representative ones.

9. Based on the above comments I am not convinced that changes in ROS accumulation link provide any evidence for their contribution to mitochondrial and chloroplast communication as discussed by the authors. The data demonstrate that the lack of the mitochondrial PPR protein SOP10 rescues the albino phenotype. However, the conclusion that this involves chloroplast-to-mitochondrion (CTM) communication that regulates cellular ROS levels at low temperatures is far too general to be accurate. As presented, the discussion section is weak and uninformative. It does not provide a critical evaluation of the data or the mechanisms involved and that the overexpression of SOD protects against the senescence -induced accumulation of hydrogen peroxide.

Response: Thank you for the comments. Our previous work showed that a dysfunction in chloroplast results in ROS accumulation and albino phenotype under low temperatures (Wang et al., 2022). Loss of function in a mitochondrial

protein SOP10 suppressed the phenotype caused by chloroplast mutation (*ospus1*). This is very similar to the previous report on *Arabidopsis mod1* (*mosaic cell death*) mutant, in which a deficiency in fatty acid biosynthesis in chloroplasts results in mtROS accumulation and cell death, which are suppressed by a mutation in a mitochondrial protein, suggesting CTM communication in *mod1*-mediated programmed cell death (Wu et al., 2015). Further study revealed malate mediated CTM communication regulating programmed cell death: the deficiency in *mod1* chloroplasts produced elevated malate in chloroplasts. The malate is then transported into mitochondria and oxidized to generate NADH, which serves as the electron donor for the mtETC to generate ROS, triggering programmed cell death. Mutations in any component of the malate shuttle pathway between chloroplasts to mitochondria block the formation of mtROS and suppress *mod1* phenotypes (Zhao et al., 2018). One of the main authors of this manuscript (Dr. Lilan Luo) is also the major contributor to the work in *Arabidopsis*.

To test whether the malate shuttle is involved in the *ospus1* albino phenotype, we created loss-of-function mutants of a malate shuttle component in the *ospus1* background (Fig. 1 for reviewers only). Malate dehydrogenases (MDHs) and malate transporters in chloroplasts and mitochondria compose the malate shuttle (Fig. 1a for reviewers only). The rice genome encodes 12 and 11 annotated MDHs and malate transporters/translocators, respectively (Heng et al., 2018; Liu et al., 2017), but only two MDHs and two malate transporters are experimentally identified (Teng et al., 2019) (Tables 1 and 2 for reviewers only). OsMDH1 (LOC_Os01g61380) is the solely identified plastid MDH, whose knockout mutants show drastically decreased MDH activity in chloroplast protein extracts (Nan et al., 2020). We generated *ospus1 osmdh1* mutant plants (Fig. 2 for reviewers only), and found it rescued the *ospus1* phenotype (Fig. 1b for review only).

We then investigated the subcellular localization of the 10 annotated MDHs that were not reported previously, and found two of them are mitochondria-localized. We tried to knock out the two mitochondria-localized putative MDHs (OsMDH2.1/LOC_Os02g01510, and OsMDH6.1/LOC_Os06g01590) in *ospus1* background, however, we did not obtain the *mdh* double knockouts in *ospus1*. The data indicate a possibility that communication between these two organelles is involved in the albino formation in *ospus1*. Therefore, we discussed such a possibility and its potential biological significance in our revised manuscript. (Lines 290–313)

10. The authors present data showing that the levels of glutathione and CAT and POD activities were also significantly higher in the transformed *ospus1-1* plants than other lines. The data for enzyme activity are expressed as units per mg fresh weight but the authors do not provide the definition of a Unit in terms of activity. This information should be provided. Similarly, the authors only provide data on reduced glutathione (GSH). It would be more informative to provide data on both GSH and glutathione disulphide. This would allow a more accurate understanding of changes in the total glutathione pool and its oxidation state.

Response: Thank you for the comment. Since the albino phenotype of *ospus1* is consistent with the NBT but not DAB staining pattern (Fig. 4a, b in the revised manuscript), the albino phenotype was likely related to the accumulated $O_2^{\cdot-}$ in the leaf blade. Therefore, assessing the hydrogen peroxide scavenging system may not be helpful for understanding the albino mechanism. We therefore removed the data

(original Fig. 4c) regarding hydrogen peroxide scavenging system.

Reviewer #2 (Remarks to the Author):

Comments on „Chloroplast-to-mitochondrion communication regulates reactive oxygen species production and cold responses in rice“ by Zu et al., submitted to Nat. Comm.

The authors build upon their earlier work, published in New Phytologist last year, characterizing a cold-sensitive, albino rice mutant affected in a chloroplast pseudouridine synthase OsPUS1, which is necessary for cp rRNA maturation at lowered temperatures of 22° vs. 28°C. The cold-sensitive PUS1 phenotype is associated with increased ROS (reactive oxygen species) accumulation. Screening for genetic suppressors in an EMS-mutagenized population they identify SOP10, encoding a mitochondrial pentatricopeptide repeat (PPR) protein. SOP10 suppresses the cold-sensitive phenotype but does not reconstitute rRNA processing in chloroplasts. Instead, SOP10 turns out to affect splicing of the respective first introns in mitochondrial *nad4* and *nad5* transcripts resulting in reduced complex I activities in the SOP10 mutants. Expressing different SODs (superoxide dismutases) can compensate the cold-sensitive phenotype and independently mutating SOP10 in three different and widely used rice *indica* varieties likewise lowers cold-sensitivity and ROS levels.

The manuscript summarizes solid experimental work and is clearly written. However, in my opinion the paper suffers from overstatements assuming “regulation” and intracellular “communication”, already starting with the paper’s title. As in similar examples in the recent scientific literature such terms often appear overused to “sell” the respective stories (see e.g., the last two sentences of the abstract). Isn’t it simply that mutation of SOP10 causes a moderate defect of mitochondrial complex 1, which results in lower ROS levels, and which is able to ameliorate the PUS1 phenotype?

Response: We thank the reviewer for the suggestion. We changed our paper’s title to “**A mitochondrial pentatricopeptide repeat protein enhances cold tolerance by modulating mitochondrial superoxide in rice**”. In addition, we modified the sentences that might be overstated throughout the manuscript; all the revised sentences are highlighted in yellow.

Our observation is very similar to a previous report on the *Arabidopsis mod1* mutant, in which malate was identified as the chloroplast-to-mitochondria communication chemical, to regulate programmed cell death (Wu et al., 2015; Zhao et al., 2018). We knocked out plastid MDH (*OsMDH1*) in *ospus1*, and found that the *ospus1 osmdh1* suppressed the albino phenotype in *ospus1* (see Fig. 1 for reviewers only). The data indicate a possibility that the malate mediated communication between these two organelles worked in albino formation in *ospus1*. Therefore, we discussed such a possibility and its potential biological significance in our revised manuscript. (Lines 290–313) See detailed response to review 1 question 9.

In that context, also note the sentence starting in line 90 speaking of “a mutation in a mitochondrial protein”, which is simply not the case here, where we just find a moderate reduction in mature mRNA levels for *nad4* and *nad5*.

Response: We are sorry for the unclear description. In context of “a mutation in a

mitochondrial protein”, the “mitochondrial protein” is referred to the mitochondrial PPR protein SOP10.

Along these lines, a display like figure 7 postulating a messenger molecule and labeling it with a question mark is not helpful.

Response: Thank you for the suggestion. Considering that the potential communication chemical was not identified yet, we removed the messenger molecule in the revised Fig. 7.

The above comments do not mean to suggest that the results are not fine and interesting, but I feel that the claims go simply too far and are not quite of the wide general interest that would seem fully adequate for the readership of Nature Communications.

Some further comments:

1. How can a G1475A mutation result in a G492N amino acid change (line 112f. and figure 1)? The change of GGC to GAC would instead cause a G-to-D codon conversion, wouldn't it?

Response: Thank you for pointing it out and we have corrected this in the revised Fig. 1c and manuscript (see Line 109).

2. It should be mentioned that the splicing defect affects group II introns (BTW: introns would be named *nad4i461g2* and *nad5i230g2*, respectively, following a general nomenclature proposal for plant organellar introns).

Response: We thank the reviewer for the expert comment. According to the reviewer's comment, the introns are named *nad4i461g2* and *nad5i230g2* following the organelle introns nomenclature rules in the revised manuscript (see Line 157).

3. The authors offer a rather arbitrary selection of references on PPR proteins (their references 34 to 38, cited in line 148). Here, it would be especially interesting to let readers know about some of the other protein factors that already have previously been shown to affect splicing of exactly these two mitochondrial introns in model systems like *Arabidopsis* (ABO6, MISF68, OZ2), maize (DEK35, DEK55, EMP8, EMP24, EMP25, EMP602, PPR18, PPR-SMR1) or even rice (FSE5, PPR939, RL1)

Response: We appreciate the recommendation. We added references about the proteins that affect *nad4i461g2* and *nad5i230g2* splicing in *Arabidopsis*, maize, and rice, in the Results (see Lines 142–144).

4. The above is quite important in this context, and the authors may find it worth discussing whether they would expect to find phenotypes similar to the one of SOP10 for “mild” mutant alleles of those factors, too.

Response: We thank the reviewer for the suggestion. We discuss this in the revised manuscript (Lines 346–364).

5. Strangely, the authors mention PLS-type variants of PPR proteins (lines 133ff.) but then rely on a rather old/outdated PPR prediction algorithm, not taking into account modern PPR HMMer profiles. Here, the alternative modern approach (see <https://ppr.plantenergy.uwa.edu.au/>) comes to quite different conclusions on the makeup of SOP10, not suggesting 13 P-type PPRs but rather 18 PLS-type (including SS-type) PPRs and, most notably, also terminal E1 and E2 motifs that are otherwise frequently signatures of organelle RNA editing factors.

Response: Thank you for the suggestion. We used the recommended prediction algorithm (<https://ppr.plantenergy.uwa.edu.au/>) and found the SOP10 is indeed composed of 18 PLS motif with C-terminal E1 and E2 motifs. We replaced the corresponding description in the text (Line 133) and in Fig. 1d. We performed a new experiment (STS-PCR-seq) to examine whether SOP10 is involved in mitochondrial RNA editing, and the results are added in the Results (Lines 169–182) and as Fig. S6 in the revised manuscript.

6. In the above context it would be interesting to learn whether the authors have used the known PPR-RNA recognition code to make prognoses about where exactly SOP10 may bind to the *nad4* and *nad5* mRNAs.

Response: Thank you for the constructive suggestion. We used the PPR-RNA recognition code to predict the binding sequences where SOP10 may interact with *nad4* and *nad5* mRNAs. *nad5* contains 2 cis-spliced and 2 trans-spliced introns, since the nucleotide sequences of the two trans-spliced introns are unavailable, we used the sequences including the CDS and the two cis-spliced introns (intron 1 and intron 4) for prediction, and the predicted SOP10 binding region located in *nad5* intron 1 and exon 2 mRNA. However, for *nad4*, the predicted SOP10 binding sequences reside in intron 3 of *nad4*, see detailed information in Fig. 3 for reviewers only.

Reviewer #3 (Remarks to the Author):

The manuscript by Zu et al. titled “Chloroplast-to-mitochondrion communication regulates reactive oxygen species production and cold responses in rice” describes a genetic screen to identify a suppressor of the low-temperature albino phenotype due to a mutation in *OsPUS1*, a plastid localized pseudouridine synthase coding gene involved in chloroplast biogenesis. The *ospus1* mutant has elevated ROS levels at 22°C compared to 28°C, correlating with the albino phenotype. The identified suppressor of *ospus1-1* gene SOP10 codes for a pentatricopeptide repeat (PPR) protein regulating splicing of mitochondrial genes *NAD4* and *NAD5*, leading to deficiencies in complex I and decreased ROS generation, reversing the albino phenotype at 22°C. Moreover, overexpression of SOD genes localizing to different cellular departments also reversed the albino phenotype of the *ospus1* mutant, suggesting that ROS accumulation was responsible for the phenotype. Lastly, the authors show that CRISPR/Cas9 generated single mutants in SOP10 improved the cold tolerance potential of three cold sensitive rice cultivars without a yield penalty.

This manuscript is a follow-up of their 2022 paper on identifying *OsPUS1* as a pseudouridine synthase required for proper chloroplast function at mild chilling temperatures. It presents an interesting story showing evidence for chloroplast-to-mitochondrion communication, which appears to be required under abiotic stress conditions to maintain ROS homeostasis. The manuscript is well written, clear and concise, and the interpretations fit the results. While it was previously shown that PPR proteins regulate RNA metabolism of plants organelles required for proper organelle biosynthesis under abiotic stress conditions, this manuscript provides evidence suggesting that chloroplast-to-mitochondrion communication regulates cold responses via ROS homeostasis even at low chilling temperatures.

Response: We appreciate your positive comments. Thank you.

To clarify a few questions I have, please address the following points:

1. Lines 164-166: The statement that “[...] results suggest that SOP10 directly binds to *nad4* and *nad5* transcripts and participates in the splicing of the first intron of their mRNAs in a temperature-independent manner” is correct based on Fig. 2 and Fig. S4. However, it seems that only primers interrogating intron 1 were used in Fig. S4. Were the other introns investigated as well? If so, please state; if not, please qualify your statements indicating that possibly other introns are retained as well in the *sop10* mutant. There is no size marker in Fig. 2c, but it looks like the entire transcript might not be spliced, therefore, please add size markers to Fig. 2 to determine the size of the un-spliced transcript. Also, *nad5* has two splice products in 9311 and the larger one is reduced in the *sop10* mutant. Please address and explain the significance of the two splice forms and why abundance of the larger one, possibly containing un-spliced introns, is reduced.

Response: Thank you for your constructive suggestion. We performed a new experiment to test whether other introns are retained in *nad4* and *nad5*. The data showed that only intron 1 of *nad4* and *nad5* was retained, the other introns were not affected by *sop10* mutation. We added the new data as Fig. S4 and Fig. S5 in the revised manuscript.

In addition, we repeated the experiment in Fig. 2c and added size markers to the blot. We found that *nad5* has only one spliced product corresponding the intact mature *nad5* transcript (about 2 kb) in 9311, the absence of the intron1-retained *nad5* spliced form in the blot may be due to its low abundance. We replaced Fig. 2c in the revised version.

2. Fig. 6: legend states “allowed to recover...for 4 days” while figure labeling says, “recover for 2-d”. Also note that “recovery” would be better than “recover” in the figure labeling.

Response: Thank you for pointing it out and for the suggestion. We corrected the figure labeling as “Recovery for 4 d” in the revised manuscript (Line 768).

Reference:

- Heng, Y., Wu, C., Long, Y., Luo, S., Ma, J., Chen, J., Liu, J., Zhang, H., Ren, Y., Wang, M., et al. (2018). OsALMT7 Maintains Panicle Size and Grain Yield in Rice by Mediating Malate Transport. *Plant Cell* *30*, 889-906. 10.1105/tpc.17.00998.
- Liu, J., Zhou, M., Delhaize, E., and Ryan, P.R. (2017). Altered Expression of a Malate-Permeable Anion Channel, OsALMT4, Disrupts Mineral Nutrition *Plant physiology* *175*, 1745-1759. 10.1104/pp.17.01142.
- MacMillan-Crow, L.A., and Crow, J.P. (2011). Does more MnSOD mean more hydrogen peroxide? *Anticancer Agents Med Chem* *11*, 178-180. 10.2174/187152011795255939.
- Nan, N., Wang, J., Shi, Y., Qian, Y., Jiang, L., Huang, S., Liu, Y., Wu, Y., Liu, B., and Xu, Z.-Y. (2020). Rice plastidial NAD-dependent malate dehydrogenase 1 negatively regulates salt stress response by reducing the vitamin B6 content. *Plant Biotechnol J* *18*, 172-184. 10.1111/pbi.13184.
- Teng, X., Zhong, M., Zhu, X., Wang, C., Ren, Y., Wang, Y., Zhang, H., Jiang, L., Wang, D., Hao, Y., et al. (2019). FLOURY ENDOSPERM16 encoding a NAD-dependent cytosolic malate dehydrogenase plays an important role in starch synthesis and seed development in rice. *Plant Biotechnol J* *17*, 1914-1927. 10.1111/pbi.13108.
- Wang, Z., Sun, J., Zu, X., Gong, J., Deng, H., Hang, R., Zhang, X., Liu, C., Deng, X., Luo, L., et al., (2022). P pseudouridylation of chloroplast ribosomal RNA contributes to low temperature acclimation in rice. *New Phytologist* *236*, 1708-1720. 10.1111/nph.18479
- Wu, J., Sun, Y., Zhao, Y., Zhang, J., Luo, L., Li, M., Wang, J., Yu, H., Liu, G., Yang, L., et al. (2015).

Deficient plastidic fatty acid synthesis triggers cell death by modulating mitochondrial reactive oxygen species. *Cell Res* 25, 621–633. 10.1038/cr.2015.46.

Zhao, Y., Luo, L., Xu, J., Xin, P., Guo, H., Wu, J., Bai, L., Wang, G., Chu, J., Zuo, J., et al. (2018). Malate transported from chloroplast to mitochondrion triggers production of ROS and PCD in *Arabidopsis thaliana*. *Cell Res* 28, 448–461. 10.1038/s41422-018-0024-8.

Figures and Tables for Reviewers only

Fig. 1 (for reviewers only) Mutation in chloroplast MDH suppressed *ospus1* phenotype.
a Key components of the malate shuttle between chloroplasts and mitochondria.
b Phenotype of 9311, *ospus1-1*, *ospus1-1 sop10-1*, and *ospus1-1 osmdh1^{CR}* at 22 °C.

Fig. 2 (for reviewers only) Genotype of the *osmdh1* mutants created by CRISPR-Cas9 technology.

The sgRNA sequences of CRISPR targeting sites on *OsMDH1* gDNA are highlighted in yellow. The predicted protein length of the *osmdh1*^{CR} mutants are shown on the right. Genotype and mutant pattern were confirmed by the Sanger sequencing.

Repeat	1	2	3	4	5	6	7	8	9	10	11	12	13	14	15	16	17	18	19	20
5 th AA	N	N	N	N	T	S	N	N	S	N	A	S	N	A	S	N	I	T	E	V
Last AA	N	N	N	N	D	D	T	N	S	D	D	D	T	N	N	D	L	D	N	K
Base preference	C>U	C>U	A>>U	C>UG>A>UG>>CC>U	C>U	C>U	A	U>C>G	G	G>>C	C>U	A>>U	A	U>C>G	?	G>A>U	?	?		
nad4	C	C	A	U	A	G	C	U	U	A	U	G	G	U	U	A	G	A	G	C
intron3	C	U	A	U	U	G	C	C	A	C	G	C	C	G	A	C	C	A	U	C
nad5	C	U	U	U	U	G	C	C	A	G	G	C	C	U	G	U	U	C	U	U
intron1																				
nad5	U	U	U	U	U	U	U	A	U	G	C	U	A	A	U	G	U	U	G	G
exon1																				

Fig. 3 (for reviewers only) Binding predictions for the SOP10 proteins on the *nad4* and *nad5* RNAs (referred to Barkan et al., 2012; Yin et al., 2013; and Gully et al., 2015). Each PPR code, which is composed of the 5th and the last amino acids of each PPR repeat, is indicated. Nucleotides matching the amino acid combination are indicated in red. “?” indicates an unidentified nucleotide.

Table 1. (for reviewers only) Summary of candidate malate dehydrogenase genes in the rice genome.

Gene	Locus	Annotation	Reference	Construct	Verified Subcellular localization
OsMDH1	LOC_Os01g61380	lactate/malate dehydrogenase	Plant Biotech. J. 2020, 18:172–184	OsMDH1	Chloroplast
OsMDH1.2	LOC_Os01g46070	lactate/malate dehydrogenase	not identified	XF4318	cytoplasm
OsMDH2.1	LOC_Os02g01510	lactate/malate dehydrogenase	not identified	XF4325	Mitochondria
OsMDH3.1	LOC_Os03g56280	lactate/malate dehydrogenase	not identified	XF4319	cytoplasm
OsMDH4.1	LOC_Os04g46560	lactate/malate dehydrogenase	not identified	XF4326	Nucleus
OsMDH5.1	LOC_Os05g49880	lactate/malate dehydrogenase	not identified	XF4320	cytoplasm
OsMDH6.1	LOC_Os06g01590	lactate/malate dehydrogenase	not identified	XF4327	Mitochondria
OsMDH7.1	LOC_Os07g43700	lactate/malate dehydrogenase	not identified	XF4321	Peroxisome
OsMDH8.1	LOC_Os08g33720	lactate/malate dehydrogenase	not identified	XF4322	cytoplasm
OsMDH8.2	LOC_Os08g44810	lactate/malate dehydrogenase	not identified	XF4324	cytoplasm
OsMDH10.1 (OsFLO16)	LOC_Os10g33800	lactate/malate dehydrogenase	Plant Biotech. J. 2019, 17:1914–1927	OsFLO16	Cytoplasmic
OsMDH12.1	LOC_Os12g43630	lactate/malate dehydrogenase	not identified	XF4323	cytoplasm

Table 2. (for reviewers only) Summary of candidate malate transporter genes in the rice genome.

Gene	Locus	Annotation	Reference	Construct	Verified Subcellular localization
OsALMT1	LOC_Os04g34010	aluminum-activated malate transporter	not identified	NA	Plasma membrane
OsALMT2	LOC_Os10g42180	aluminum-activated malate transporter	not identified	XF4315	Mitochondria
OsALMT3	LOC_Os02g49790	aluminum-activated malate transporter	not identified	NA	NA
OsALMT4	LOC_Os01g12210	aluminum-activated malate transporter	Plant Physiol. 2017, 175(4):1745-1759		Plasma membrane
OsALMT5	LOC_Os01g53570	aluminum-activated malate transporter	not identified	NA	NA
OsALMT6	LOC_Os06g15779	aluminum-activated malate transporter	not identified	NA	NA
OsALMT7	LOC_Os02g45160	aluminum-activated malate transporter	Plant Cell. 2018, 30(4):889-906		Plasma membrane
OsALMT8	LOC_Os04g47930	aluminum-activated malate transporter	not identified	NA	NA
OsALMT9	LOC_Os06g22600	aluminum-activated malate transporter	not identified	NA	NA
OsOMT/DiT1	LOC_Os11g24450	mitochondrial 2-oxoglutarate/malate translocator	not identified	XF4316	Mitochondria
OsOMT1/DIT2	LOC_Os12g33080	2-oxoglutarate/malate translocator	not identified	XF4317	Mitochondria (chloroplast?)

Revised Figures in the manuscript

Revised Fig. 1c, d **c** Schematic diagram of *SOP10* gene. The black triangle indicates the mutation site (G1475A; G492D) in *ospus1-1 sop10-1*. **d** Protein structure of OsSOP10 in rice, the PPR motifs were shown in different colored box.

Revised Fig. 2c **c** Northern blot analysis to detect the *nad4* and *nad5* transcripts. Total RNA was extracted from seedlings of 9311, *ospus1-1*, four suppressors of *ospus1-1*, and *comp* plants grown at 28°C and 22°C. The specific fragments of *nad4* and *nad5* used as the probes. Red triangles indicate the intron retention in *nad4*. Methylene blue staining (MB stain) is shown as a loading control.

Revised Fig. 4b Schematic drawing of relationship between albino phenotype and ROS accumulation pattern in *ospus1-1* at 22°C. The albino occurred in the whole leaf blade which is consistent with $O_2^{\bullet-}$ accumulation pattern.

Revised Fig. 5c Accumulated ROS in *ospus1-1* was suppressed by overexpressing SOD-encoding genes. The accumulation of $O_2^{\bullet-}$ and H_2O_2 in the leaf of rice grown at 22°C in (b) were assessed by NBT and DAB staining, respectively. The numbers on top of each photo show how often this representative staining pattern occurred, relative to total examined pictures. Scale bars, 1 cm.

Revised Fig. 6 | Assessment of cold stress tolerance of *sop10* single mutant plants at seedling stage. **a** Seedlings of LKZ1, ZJZ17, ZZ35, and their two corresponding *sop10^{CR}* mutants were grown at 28°C for 2 weeks, shifted to 8°C for 2 days, and allowed to recover at 28°C for 4 days. Scale bars, 8 cm. **b** Survival rates of LKZ1, ZJZ17, ZZ35, and their *sop10^{CR}* seedlings after cold treatment in (a). Values are means \pm S.D. ($n = 3$ biological replicates). The asterisks represent significant differences between mutants and wild type grown under the same conditions, determined by Student's *t*-test (two-tailed). *, $P < 0.05$; **, $P < 0.01$; *ns*, no significant difference. **c** NBT and DAB staining were used to assess the accumulation of $O_2^{\cdot-}$ and H_2O_2 in LKZ1, ZJZ17, ZZ35, and their *sop10^{CR}* mutant seedlings before and after cold treatment in (a). The numbers on top of each photo show how often this representative staining pattern occurred, relative to total examined pictures. Scale bars, 1 cm. **d** In-gel assay of NADH oxidase capacity of LKZ1, ZJZ17, ZZ35, and their *sop10^{CR}* mutants. Dihydropyridine dehydrogenase activity was used as a loading control. The red triangle indicates mitochondrial complex I, and the black triangle indicates dihydropyridine.

Revised Fig. 7 | A proposed model of albino formation mediated by *ospus1* under low temperature. (Left) In wild-type 9311, PUS1 is upregulated and catalyzes the pseudouridylation of pre-rRNAs under cold conditions, enabling normal chloroplast ribosome assembly and translation, thus maintaining homeostasis in chloroplast metabolism. **(Middle)** A deficiency in chloroplast pre-rRNA pseudouridylation in *ospus1* mutants results in aberrant chloroplast ribosomes and defects in plastid translation, which then leads to an imbalance of chloroplast metabolism. The disruption of chloroplast homeostasis generated a chemical/signal that was transmitted into mitochondrion and induced $O_2^{\bullet-}$ overproduction from the mitochondrial ETC (mETC) complex I. The mitochondrial $O_2^{\bullet-}$ then caused the albino phenotype in *ospus1* under low temperatures, by oxidizing target proteins (Oxi-PTMs), which remain to be identified. **(Right)** Mutations in a mitochondrial PPR protein (SOP10) that directly binds to *nad4* and *nad5* transcripts and regulates splicing of their first exons, decreases mETC complex I capacity and $O_2^{\bullet-}$ accumulation, and suppresses the albino phenotype of *ospus1*, enhances rice cold tolerance in wild-type 9311.

Supplemental Figures in the manuscript

Fig. S4 The first intron of *nad4* is retained in *ospus1-1 sop10-1* plants. **a** Diagram of *nad4* gene structure. Positions of the primers used for PCR are indicated by black arrows. **b** Intron retention occurred in *nad4* transcripts in suppressors of *ospus1-1* (*ospus1-1 sop10-1* and three *ospus1-1 sop10^{CR}* lines) at 28°C and 22°C by RT-PCR. The purple triangle represents the intron-retained *nad4* transcript. **c** The first intron of *nad4* was retained in *ospus1-1 sop10-1* plants. Sanger sequencing of RT-PCR products amplified in **b**.

Fig. S5 The first intron of *nad5* is retained in *ospus1-1 sop10-1* plants. **a** Diagram of *nad5* gene structure. Positions of the primers used for PCR are indicated by black arrows. **b** Intron retention occurred in *nad5* transcripts in suppressors of *ospus1-1* (*ospus1-1 sop10-1* and three *ospus1-1 sop10^{CR}* lines) at 28°C and 22°C by RT-PCR. The triangle represents the intron-retained *nad5* transcript. **c** The first intron of *nad5* was retained in *ospus1-1 sop10-1* plants. Sanger sequencing of RT-PCR products amplified in **b**.

Fig. S6| RNA editing at five mitochondrial sites is defective in the *sop10-2* mutant. **a** The editing efficiency is shown by T/(T+C)% in the WT and *sop10-2*. The relevant raw data are reported in Supplemental Table 3. Two independent RNA samples were used to analyze the editing by STS-PCR-seq. The bars show the mean \pm S.D. ($n = 2$ biological replicates). **b** The editing sites were confirmed by Sanger sequencing using independent RNA samples from those two used in **a**. The RNA editing sites and the resulting amino acid changes are illustrated below.

REVIEWERS' COMMENTS

Reviewer #1 (Remarks to the Author):

Your manuscript has been appropriately revised to address all the points raised.

Reviewer #2 (Remarks to the Author):

Revision of manuscript NCOMMS-23-02347A by Zu and colleagues, submitted to Nature Communications

The manuscript has been revised quite extensively in response to comments by three reviewers focusing on different subjects ranging from ROS accumulation and metabolism to details of molecular genetics concerning mitochondrial gene expression and PPR protein makeup. Judging on the latter part, concerning my core expertise, the paper has now improved in content and quality with a better molecular characterization of the SOP10 protein and, as an immediate consequence, the additional finding of RNA editing defects on top of splicing defects (new section, lines 169-182).

Accordingly, corresponding text has been added. The added discussion of splicing factors (lines 346 ff.) needs some re-checking, however. For example, intron nad5i392g2 is not present in flowering plants but unique to Lycopodiales, the (trans-splicing) nad5 “intron 2” of angiosperms in question is nad5i1455g2. Also, it may be worth noting and emphasizing that MISF68 and ABO6, like SOP10 presented here, also affect both nad4i461g2 and nad5i230g2 (and other introns) at the same time. Rice PPR939 affects nad5 introns 1, 2 and 3.

One of the major changes not addressed in the authors' response letter is that FIVE more co-authors from TWO more laboratories have now been added. In the paper, these contributions are now given for four of the new co-authors simply as “Y. W., X. Q., C. C. and B. T. helped to write the paper.” This evidently is an editorial issue but from my point of view the addition of 5 more co-authors would need to be better elaborated and justified.

Reviewer #3 (Remarks to the Author):

This is a resubmission of the Zu et al. manuscript “Chloroplast-to-mitochondrion communication regulates reactive oxygen species production and cold responses in rice”, now titled “A mitochondrial pentatricopeptide repeat protein enhances cold tolerance by modulating mitochondrial superoxide in rice”. The authors addressed all my concerns and comments and I have further suggestions for improving the manuscript.

Dear Dr. Xu and reviewers:

Thank you for your letter and for the reviewers' comments concerning our manuscript (NCOMMS-23-02347A). We have read the comments and have made corresponding revisions. Revised portions are highlighted in yellow in the paper.

Our responses to the reviewers' comments are as follows:

Reviewer #1 (Remarks to the Author):

Your manuscript has been appropriately revised to address all the points raised.

Response: We appreciate your positive comments. Thank you.

Reviewer #2 (Remarks to the Author):

Revision of manuscript NCOMMS-23-02347A by Zu and colleagues, submitted to Nature Communications

The manuscript has been revised quite extensively in response to comments by three reviewers focusing on different subjects ranging from ROS accumulation and metabolism to details of molecular genetics concerning mitochondrial gene expression and PPR protein makeup. Judging on the latter part, concerning my core expertise, the paper has now improved in content and quality with a better molecular characterization of the SOP10 protein and, as an immediate consequence, the additional finding of RNA editing defects on top of splicing defects (new section, lines 169-182). Accordingly, corresponding text has been added.

The added discussion of splicing factors (lines 346 ff.) needs some re-checking, however. For example, intron *nad5i392g2* is not present in flowering plants but unique to Lycopodiales, the (trans-splicing) *nad5* "intron 2" of angiosperms in question is *nad5i1455g2*.

Response: We thank the reviewer for the expert comments. We corrected the *nad5* intron 2 *nad5i392g2* into *nad5i1455g2* (see Line 347).

Also, it may be worth noting and emphasizing that MISF68 and ABO6, like SOP10 presented here, also affect both *nad4i461g2* and *nad5i230g2* (and other introns) at the same time. Rice PPR939 affects *nad5* introns 1, 2 and 3.

Response: Thanks. We noted and emphasized that the role of Arabidopsis MISF68 and ABO6 in splicing of both *nad4i461g2* and *nad5i230g2* (Line 348), and mentioned that rice PPR939 affects *nad5* introns 2, 3 except for intron 1 (Line 352).

One of the major changes not addressed in the authors' response letter is that FIVE more co-authors from TWO more laboratories have now been added. In the paper, these contributions are now given for four of the new co-authors simply as "Y. W., X. Q., C. C. and B. T. helped to write the paper." This evidently is an editorial issue but from my point of view the addition of 5 more co-authors would need to be better elaborated and justified.

Response: Thank you. One of the added author Chunhui Xu performed the data analysis of the STS-PCR-seq, and we included this information in the manuscript (Line 681).

Reviewer #3 (Remarks to the Author):

This is a resubmission of the Zu et al. manuscript "Chloroplast-to-mitochondrion communication regulates reactive oxygen species production and cold responses in

rice" , now titled "A mitochondrial pentatricopeptide repeat protein enhances cold tolerance by modulating mitochondrial superoxide in rice" . The authors addressed all my concerns and comments and I have further suggestions for improving the manuscript.

Response: We thank the reviewer for the positive comment.